# REVISITING LAYER-WISE SAMPLING IN FAST TRAINING FOR GRAPH CONVOLUTIONAL NETWORKS

## ABSTRACT

To accelerate the training of graph convolutional networks (GCNs), many sampling-based methods have been developed for approximating the embedding aggregation. Among them, a layer-wise approach recursively performs importance sampling to select neighbors jointly for existing nodes in each layer. This paper revisits the approach from a matrix approximation perspective. We identify two issues in the existing layer-wise sampling methods: sub-optimal sampling probabilities and the approximation bias induced by sampling without replacement. We thus propose remedies to address these issues. The improvements are demonstrated by extensive analyses and experiments on common benchmarks.

## 1 INTRODUCTION

Graph Convolutional Networks (Kipf & Welling, 2017) are popular methods for learning the representation of nodes. However, it is computationally challenging to train a GCN over large-scale graphs due to the inter-dependence of nodes in a graph. In the mini-batch training for an L-layer GCN, the computation of embeddings involves not only the batch nodes but also batch nodes' $L$-hop neighbors, which is known as the phenomenon of "neighbor explosion" (Zeng et al., 2019) or "neighbor expansion" (Chen et al., 2018a; Huang et al., 2018). To alleviate such a computation issue for large graphs, sampling-based methods are proposed to accelerate the training and reduce the memory cost. These approaches can be categorized as node-wise sampling approaches (Hamilton et al., 2017; Chen et al., 2018a), subgraph sampling approaches (Zeng et al., 2019; Chiang et al., 2019; Cong et al., 2020), and layer-wise sampling approaches (Chen et al., 2018b; Huang et al., 2018; Zou et al., 2019). We focus on layer-wise sampling in this work, which enjoys the efficiency and variance reduction by sampling columns of renormalized Laplacian matrix in each layer.

This paper is a study of the existing sampling schemes in layer-wise sampling methods. We identify two potential drawbacks in the common practice for layer-wise sampling (Chen et al., 2018b; Zou et al., 2019). First, the current sampling probabilities used are sub-optimal since a core assumption in FastGCN and LADIES does not hold in many common graph benchmarks, such as Reddit (Hamilton et al., 2017) and OGB(Hu et al., 2020). Secondly, the previous implementations of the layer-wise sampling methods slightly deviate from their theoretical results, and introduce bias in the estimation due to the usage of sampling without replacement. Realizing the two issues, we accordingly propose the remedies with new sampling probabilities and a debiasing algorithm. The improvements of the proposed methods are demonstrated by extensive experiments on evaluating the matrix approximation error and prediction accuracy on large-scale benchmarks along with some theoretical analyses.

To the best of our knowledge, our result is the first to recognize and resolve the issues with the default assumption and the practical implementation of layer-wise sampling methods for GCN. Once these sub-optimal practices are addressed, we observe that the GCN models consistently converge faster in training and usually enjoy a higher prediction accuracy. We believe the proposed methods can more generally improve the training for GCN as well, e.g., the same strategy can allow node-wise sampling methods to adopt sampling without replacement and further improve the approximation accuracy. Moreover, our discussion on the bias induced by sampling without replacement is not limited to GCN, and the debiasing algorithm we develop can contribute to other sampling-based machine learning models beyond layer-wise sampling.

## 1.1 BACKGROUND AND RELATED WORK

**GCN** Graph Convolutional Networks (GCNs, Kipf & Welling (2017)) effectively incorporate the technique of convolution filter into the graph domain (Wu et al., 2020; Bronstein et al., 2017). Viewed as an approximation for the spectral graph convolutions (Bruna et al., 2014; Defferrard et al., 2016), GCN has achieved great success in learning tasks such as node classification and link prediction, with applications ranging from recommender systems (Ying et al., 2018), traffic prediction (Cui et al., 2019; Rahimi et al., 2018), and knowledge graphs (Schlichtkrull et al., 2018).

**Sampling Based GCN Training** To name a few of sampling schemes, GraphSAGE (Hamilton et al., 2017) first introduces the "node-wise" neighbor sampling scheme, where a fixed number of neighbors are uniformly and independently sampled for each node in every layer. To reduce variance in node-wise sampling training, VR-GCN (Chen et al., 2018a) applies a control variate approach with historical activation. Instead of sampling for each node, "layer-wise" sampling is a more efficient approach: joint sampling scheme for all the existing nodes in each layer so that these nodes can share the sampled neighboring node. FastGCN (Chen et al., 2018b) first introduces this scheme with importance sampling. AS-GCN (Huang et al., 2018) proposes an alternative sampling probability for layer-wise sampling by approximating the hidden layer in the sampling procedure. Then Zou et al. (2019) propose a layer-dependent importance sampling scheme (LADIES) to further reduce the variance in training. This alleviates the issue of empty rows in sampled adjacency matrix for FastGCN. For the "subgraph" approach, ClusterGCN (Chiang et al., 2019) samples a dense subgraph associated with the batch nodes by graph clustering algorithm; GraphSAINT (Zeng et al., 2019) introduces normalization and variance reduction in subgraph sampling.

## 2 NOTATIONS AND PRELIMINARIES

### 2.1 GRAPH CONVOLUTIONAL NETWORKS

The GCN architecture for semi-supervised node classification is introduced by Kipf & Welling (2017). Suppose we have an undirected graph $\mathcal{G} = (\mathcal{V}, \mathcal{E})$, where $\mathcal{V}$ is the set of $n$ nodes and $\mathcal{E}$ is the set of $E$ edges. Denote node $i$ in $\mathcal{V}$ as $v_i$, where $i \in [n]$ is the index of nodes in the graph and $[n]$ denotes the set $\{1, 2, ..., n\}$. Each node $v_i \in \mathcal{V}$ is associated with a feature vector $x_i \in \mathbb{R}^p$ and a label vector $y_i \in \mathbb{R}^q$. Though we can observe the feature of every node in $\mathcal{V}$ and every edge in $\mathcal{E}$, i.e. the $n \times n$ adjacency matrix $A$, we are only able to observe the label of partial nodes $\mathcal{V}_{train}$, satisfying $\mathcal{V}_{train} \subset \mathcal{V}$. Thus, we need to predict the labels for the rest nodes in $\mathcal{V} \backslash \mathcal{V}_{train}$ and it becomes a semi-supervised learning task. A graph convolution layer is defined as:

$$\boldsymbol{Z}^{(l+1)} = \boldsymbol{P}\boldsymbol{H}^{(l)}\boldsymbol{W}^{(l)}, \quad \boldsymbol{H}^{(l)} = \sigma(\boldsymbol{Z}^{(l)}), \tag{1}$$

where $\sigma$ is an activation function and $\boldsymbol{P}$ is obtained from applying normalization to the graph adjacency matrix $\boldsymbol{A}$; $\boldsymbol{H}^{(l)}$ is the embedding matrix of the graph nodes in the $l$-th layer, and $\boldsymbol{W}^{(l)}$ is the parameter matrix of the same layer. In particular, $\boldsymbol{H}^{(0)}$ is the $n \times p$ feature matrix. For mini-batch training, the training loss for an $L$-layer GCN is defined as $\frac{1}{|\mathcal{V}_{batch}|} \sum_{v_i \in \mathcal{V}_{batch}} \ell(y_i, z_i^{(L)})$, where $\ell$ is the loss function, batch nodes $\mathcal{V}_{batch}$ is a subset of $\mathcal{V}_{train}$ at each iteration. $z_i^{(L)}$ is the $i$-th row in $\boldsymbol{Z}^{(L)}$, $|\cdot|$ denotes the cardinality of a set.

In this paper, we set $\boldsymbol{P} = \tilde{\boldsymbol{D}}^{-1/2}(\boldsymbol{A} + \boldsymbol{I})\tilde{\boldsymbol{D}}^{-1/2}$, where $\tilde{D}$ is a diagonal matrix with $\boldsymbol{D}_{ii} = 1 + \sum_i \boldsymbol{A}_{ij}$. The matrix $\boldsymbol{P}$ is constructed as a *renormalized Laplacian matrix* to help alleviate overfitting and exploding/vanishing gradients issues (Kipf & Welling, 2017), which is used by Kipf & Welling (2017); Chen et al. (2018a); Cong et al. (2020).

### 2.2 LAYER-WISE SAMPLING

To address "neighbor explosion" issue for graph neural networks, sampling methods are integrated into the stochastic training. Motivated by the idea to approximate the matrix $\boldsymbol{P}\boldsymbol{H}^{(l)}$ in (1), FastGCN (Chen et al., 2018b) applies an importance-sampling-based strategy. Instead of individually sampling neighbors for each node in the $l$-th layer, they sample a set of $s$ neighbors $\mathcal{S}^{(l)}$ from $\mathcal{V}$ with importance sampling probability $p_i$, where $p_i \propto \sum_{j=1}^n \boldsymbol{P}_{ji}^2$ and $\sum_i p_i = 1$. For the $(l-1)$-th layer,

Figure 1: Compare matrix approximation error for layer-wise sampling methods. The matrix approximation error is measured by $\|\tilde{Z}_{batch}^{(1)} - \tilde{Z}_{sampling}^{(1)}\|_F$. The error curve of the original LADIES method shows an abnormal U-shape on ogbn-arxiv and ogbn-mag datasets.

they naturally set $\mathcal{V}^{(l-1)} = S^{(l)}$. LADIES (Zou et al., 2019) improves the importance sampling probability $p_i$ as

$$p_i^{(l)} \propto \sum_{v_j \in \mathcal{N}^{(l)}} \boldsymbol{P}_{ji}^2, \forall i \in [n] \tag{2}$$

where $\mathcal{N}^{(l)} = \cup_{v_i \in \mathcal{V}^{(l)}} \mathcal{N}(v_i)$ and $\sum_j p_j^{(l)} = 1$. In this case, $\mathcal{S}^{(l)}$ the nodes sampled for the $l$-th layer are guaranteed to be within the neighborhood of $\mathcal{V}^{(l)}$. The whole procedure can be concluded by a diagonal matrix $\boldsymbol{S}^{(l)} \in \mathbb{R}^{n \times n}$ and a row selection matrix $\boldsymbol{Q}^{(l)} \in \mathbb{R}^{s_l \times n}$, which are defined as

$$\boldsymbol{Q}_{k,j}^{(l)} = \begin{cases} 1, & j = i_k^{(l)} \\ 0, & \text{else} \end{cases}, \quad \boldsymbol{S}_{j,j}^{(l)} = \begin{cases} (s_l p_{i_k^{(l)}}^{(l)})^{-1}, & j = i_k^{(l)} \\ 0, & \text{else}, \end{cases}$$

where $\{i_k^{(l)}\}_{k=1}^{s_l}$ are the indices of rows selected in the $l$-th layer. The forward propagation with layer-wise sampling can thus be equivalently represented as $\tilde{\boldsymbol{Z}}^{(l+1)} = \boldsymbol{Q}^{(l+1)} \boldsymbol{P} \boldsymbol{S}^{(l)} \boldsymbol{H}^{(l)} \boldsymbol{W}^{(l)}, \boldsymbol{H}^{(l)} = (\boldsymbol{Q}^{(l)})^T \sigma(\tilde{\boldsymbol{Z}}^{(l)})$, where $\tilde{\boldsymbol{Z}}^{(l+1)}$ is the approximation of the embedding matrix for layer $l$.

## 3 EXPERIMENTAL SETUP

In advance of the formal introduction to the the issues and the corresponding remedies in Section 4 and Section 5, we state the basic setups of the main experiments and datasets as they appear multiple times across the paper. Details about GCN model training are deferred to the according sections.

**Main experiments.** To study the influence of the aforementioned issues we evaluate the matrix approximation error (c.f. Figure 1) of different methods and consider it as a new metric to reflect the performance of the sampling strategy on approximating the original mini-batch training in one-step propagation. Since the updates of parameters in the training is not involved in the simple metric above, in Section 6 we further evaluate the prediction accuracy on testing sets of both intermediate models during training and final outputs, using the metrics in Table 2.

**Benchmarks.** Empirical experiments are conducted on 5 datasets (see details at Table 2 in Appendix B): Reddit (Hamilton et al., 2017), ogbn-arxiv, ogbn-proteins, ogbn-mag and ogbn-products (Hu et al., 2020). Reddit is a traditional large graph dataset used by Chen et al. (2018b); Zou et al. (2019); Chen et al. (2018a); Cong et al. (2020); Zeng et al. (2019). Ogbn-arxiv, ogbn-proteins and ogbn-products are Open Graph Benchmarks (OGB) proposed by Hu et al. (2020). Compared to traditional datasets, our selected OGB data have larger volume (up to million-node scale) with more challenging data split. The metrics in Table 2 follow the choices of recent works and the recommendation by (Hu et al., 2020).

## 4 RECONSIDER IMPORTANCE SAMPLING PROBABILITIES

The efficiency of layer-wise sampling comes from sampling, and the choice of sampling probabilities impacts the prediction accuracy of GCNs. To minimize the variance (for the sake of notational

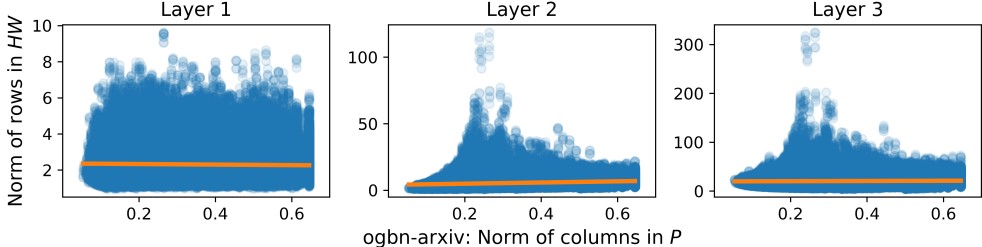

Figure 2: Fit $y^{(l)} \sim \beta_0 + \beta_1 x$ on ogbn-arxiv data. The orange solid line denotes the fitted regression line. All pairs of $(x, y)$ are collected from 5 repeated experiments and after the convergence of models. Details of experimental setups, results for other data are deferred to Appendix C.3.

brevity, from now on we omit the superscript $(l)$ when the objects are from the same layer)

$$\mathbb{E} \left\| \boldsymbol{QPSHW} - \boldsymbol{QPHW} \right\|_F^2,$$

Zou et al. (2019) show that the optimal sampling probability $p_i$ for node $i$ satisfies (also see Appendix E for a derivation from a perspective of approximate matrix multiplication)

$$p_i \propto \left\| (\boldsymbol{HW})_{[i]} \right\| \cdot \left\| \boldsymbol{QP}^{[i]} \right\|,$$

where for a matrix $\boldsymbol{A}$, $\boldsymbol{A}_{[i]}$ and $\boldsymbol{A}^{[i]}$ respectively represent the $i$-th row / column of matrix $\boldsymbol{A}$.

### 4.1 CURRENT APPROXIMATION STRATEGIES AND THEIR LIMITATIONS

The optimal sampling probabilities discussed above are usually unavailable during the mini-batch training. $\boldsymbol{W}$ and $\boldsymbol{H}$ keep changing in the training and even worse we only have access to part of the matrix $\boldsymbol{HW}$ in each batch. To approximate the optimal sampling probability, previous works develop two different strategies, 1) approximating the hidden activation $\boldsymbol{HW}$, or 2) sampling without the information from $\boldsymbol{HW}$.

A representatives of the former strategy is AS-GCN (Huang et al., 2018), which linearly approximate $\boldsymbol{HW}$ by node features. This strategy improves the accuracy of efficient GCN, while the estimation of hidden activation causes considerable overall training time, which in practice is "even longer than vanilla GCN" (Zeng et al., 2019). Another issue of this strategy is that the approximation must be updated during the training procedure. This dependence between sampling and training renders it impossible to save training time by preparing sampling results in advance of the training.

Instead, the second strategy, the theme of our paper, allows the decoupling of sampling and training, and has been adopted by FastGCN (Chen et al., 2018b) and LADIES (Zou et al., 2019). To proceed without the information from $\boldsymbol{HW}$, FastGCN (resp. LADIES) assumes $\left\| (\boldsymbol{HW})_{[i]} \right\| \propto \left\| \boldsymbol{P}^{[i]} \right\|$ (resp. $\left\| \boldsymbol{QP}^{[i]} \right\|$), and sets their sampling probabilities as $p_i \propto \left\| \boldsymbol{P}^{[i]} \right\|^2$ (resp. $\left\| \boldsymbol{QP}^{[i]} \right\|^2$), $\forall i \in [n]$. However, we find this assumption is too strong. To test it, we conduct linear regression $y^{(l)} \sim \beta_0 + \beta_1 x$ for different layers separately ($l = 1, 2, 3$), where each $y^{(l)}$ is the $\ell_2$ norm of one row in $\boldsymbol{H}^{(l)} \boldsymbol{W}^{(l)}$ and each $x$ is the norm of corresponding column in $\boldsymbol{P}$. The result of ogbn-arxiv is presented in Figure 2. For layer 1 to layer 3, the intercepts (with stand error) of the regression lines are $2.364 (\pm 0.002), 4.014 (\pm 0.008)$ and $19.722 (\pm 0.024)$ respectively, which are much larger than zero with the statistical significance; however, the according slopes $\beta_1$'s are at a smaller scale: $-0.154 (\pm 0.005), 4.468 (\pm 0.021)$ and $2.158 (\pm 0.069)$. Note that $\beta_1$ for layer 1 is even negative with statistical significance. Such evidence shows that the relation between $\left\| (HW)_{[i]} \right\|$ and $\left\| P^{[i]} \right\|$ is pretty unstable and might be even negatively correlated. This violates the assumption that $\left\| (HW)_{[i]} \right\| \propto \left\| P^{[i]} \right\|$. Similar patterns are also observed on the other datasets (See Appendix C.3).

### 4.2 PROPOSED SAMPLING PROBABILITIES

As analyzed above, the sampling probabilities in LADIES are sub-optimal due to the failure of their assumption. To address this issue, we instead admit that we have no prior knowledge of $\boldsymbol{HW}$, and

tend to assume a uniform distribution of $\|(\boldsymbol{HW})_{[i]}\|$'s. With this belief we propose the following sampling probabilities:

$$p_i \propto \|\boldsymbol{QP}^{[i]}\|, \quad \forall i \in [n]. \tag{3}$$

Compared to the sampling probabilities of LADIES in Equation (2), our proposed sampling probabilities $p_i$'s are more conservative. From a matrix approximation perspective, we rewrite the target matrix product as $\boldsymbol{QPIHW}$, and only aim to approximate the known part $\boldsymbol{QPI}$. It turns out that assuming the uniform distribution of the norms of rows in $\boldsymbol{HW}$ can help improve both the variance of the matrix approximation and the prediction accuracy of GCNs. We empirically justify our sampling probabilities in the next section and Section 6. The theoretical analysis of the new probability is presented in Section 4.4.

### 4.3 EVALUATION ON MATRIX APPROXIMATION ERROR

To further justify our proposed sampling probability, we consider the following 1-layer matrix approximation error, which evaluates the propagation approximation to the embedding aggregation of full-batch.

$$\|\tilde{\boldsymbol{Z}}_{batch}^{(1)} - \tilde{\boldsymbol{Z}}_{sampling}^{(1)}\|_F = \|\boldsymbol{Q}_{batch}\boldsymbol{PH}^{(0)}\boldsymbol{W}^{(0)} - \boldsymbol{Q}_{batch}\boldsymbol{PSH}^{(0)}\boldsymbol{W}^{(0)}\|_F,$$

where $\tilde{\boldsymbol{Z}}_{batch}^{(1)}$ and $\tilde{\boldsymbol{Z}}_{sampling}^{(1)}$ are the embedding at the bottom layer for the whole batch (using all avaiable neighbors without sampling) and for a certain sampling method; $\boldsymbol{S}$ is the sampling matrix; $\boldsymbol{Q}_{batch}$'s 0, 1 diagonal entries indicate if a node is in the batch; $\|\cdot\|_F$ denotes the Frobenius norm of a matrix. The experiments are repeated 200 times, in which we regenerate the batch nodes (shared by all sampling methods) and the sampling matrix for each method. The batch size is fixed as 512 and the numbers of sampled neighbors are $256, 512, 768, 1024, 2536$, and $2048$. $\boldsymbol{W}^{(0)}$ is fixed and inherited from the trained model reported in Section 6.

It is not surprising to observe that in Figure 1, the result of our proposed sampling probabilities (denoted as "LAIDES+flat", blue solid line) is consistently better than that of the original LADIES method (black dashed line) and of FastGCN (black solid line) on every dataset. This supports the new sampling probability in Equation (3). The discussion of the other (debiased) methods in this figure is deferred to Section 5.

### 4.4 THEORETICAL ANALYSIS OF THE NEW PROBABILITIES

Whether our choice of probabilities can outperform the previous ones depends on the distribution of the norms of rows in $\boldsymbol{HW}$. When $\|\boldsymbol{HW}_{(i)}\|$ is not proportional to the corresponding $\ell_2$ norm of column $(\boldsymbol{QP})^{(i)}$, our proposed probabilities can benefit the approximate matrix multiplication task more than the ones assuming a relation of proportionality. We find the common long-tail distribution of numbers suffices to exert the strengths of the new probabilities, which can be concluded as the following assumption:

**Assumption 1.** *To simplify the notation we denote $\boldsymbol{B} := \boldsymbol{QP}$ and $\boldsymbol{C} := \boldsymbol{HW}$, where $\boldsymbol{P}$ is an $n$-by-$n$ matrix as defined above. Let $m$ be the number of non-zero columns in $\boldsymbol{B}$, and define $C_1 := \frac{\|\boldsymbol{B}\|_F^2/m}{\left(\sum_{i=1}^n \|\boldsymbol{B}^{[i]}\|/m\right)^2} \geq 1$. There also exists a constant $C_2 \geq 1$ such that $\frac{1}{C_2}\|\boldsymbol{C}\|_F^2/n \leq \|\boldsymbol{C}_{[i]}\|^2 \leq C_2\|\boldsymbol{C}\|_F^2/n$. Assume $C_1/C_2^2 \geq 1$.*

With the assumption above, we show the variance of the approximation with our proposed probabilities is smaller than the variance of LADIES by the following lemma.

**Lemma 1.** *We denote the sampling matrix with our probabilities in Equation (3) as $\boldsymbol{S}_1$, and denote the sampling matrix with probabilities of LADIES in Equation (2) as $\boldsymbol{S}_0$. If Assumption 1 holds, then we have*

$$\mathbb{E}\|\boldsymbol{BS}_1\boldsymbol{C} - \boldsymbol{BC}\|_F^2 \leq \mathbb{E}\|\boldsymbol{BS}_0\boldsymbol{C} - \boldsymbol{BC}\|_F^2.$$

The proof is provided in Appendix F.

**Remark.** Assumption 1 is related to the uniformity in the distributions of $\|\boldsymbol{B}^{[i]}\|$'s and $\|\boldsymbol{C}_{[i]}\|$'s. We tentatively discuss the implication of the assumption in Appendix F. We remark the assumption

indicates it is unrealistic that the new probabilities can outperform the ones in LADIES, as distributions of datasets can vary. Nevertheless, as shown in Section 6 it can be an effective attempt to improve the prediction accuracy of LADIES by simply adopting the conservative sampling scheme.

## 5 DEBIAS SAMPLING WITHOUT REPLACEMENT

The theoretical results derived by Chen et al. (2018b); Zou et al. (2019) guarantee their methods are unbiased approximations of the embedding aggregations in GCN training, with an independent and identically distributed (i.i.d.) sub-sample assumption. However, biases are induced in the implementations due to the usage of sampling without replacement. As illustrated by the U-shape curves of ogbn-arxiv and ogbn-mag datasets in Figure 1, in the long run the matrix approximation error of LADIES even increases with the number of sub-samples. The trend indicates the existence of biases since given an unbiased estimation, the matrix approximation error is expected to decrease with more sub-samples used. In particular, such bias is more significant in sparse graphs with small average degrees. In the following subsections, we analyze the implementation of sampling without replacement in LADIES to demonstrate the origin of the bias, and propose a new correction method to debias the estimation while keep using sampling without replacement.

**Remark.** We insist on the usage of sampling without replacement because it does diminish the variance of the estimation. This practice indeed works for node-wise sampling methods adopting simple random sampling (SRS, sampling with all-equal probabilities). Specifically, assuming $m$ items are uniformly sampled without replacement from a population of size $n$, using the same formula for estimation as in sampling with replacement is still unbiased and enjoys the reduction of variance by a finite population correction (FPC) factor $\frac{n-m}{n-1}$ (Lohr, 2019, Section 2.3).

### 5.1 WEIGHTED RANDOM SAMPLING WITHOUT REPLACEMENT

We first introduce the implementation and necessary notations of non-uniform sampling without replacement applied in FastGCN and LADIES. Given a set $V = [n]$ representing the indices of $n$ items (matrices in the layer-wise sampling setting) $\{X_i\}_{i=1}^n$ and associated sampling probabilities $\{p_i\}_{i=1}^n$, the algorithm can be stated as a sequential procedure named as weighted random sampling (WRS) (Efraimidis & Spirakis, 2006, Algorithm D). Specifically, aside from the set $S_k$ of $k$ previously sampled indices ($0 \leq k \leq m - 1, S_0 := \emptyset$), the $k + 1$-th random index $I_{k+1}$ is sampled from the set $V - S_k$ of the rest $n - k$ indices with probabilities

$$
\begin{aligned}
p_i^{(0)} &:= p_i, && \forall i \in V = [n], \\
p_i^{(k)} &:= \frac{p_i}{\sum_{j \in V - S_k} p_j}, && \forall k \in [m-1], i \in V - S_k.
\end{aligned}
$$

In the implementation of previous layer-wise sampling methods, the formula used to estimate the target $\sum_{i=1}^n X_i$ is displayed (after adapted to the notations in this subsection) as follows:

$$
\frac{1}{m} \sum_{k=0}^{m-1} X_{I_{k+1}} / p_{I_{k+1}}, \tag{4}
$$

which is derived under the sampling with replacement assumption. With the notations introduced, we are now able to analyze the effect of directly using Equation (4) while the WRS algorithm is performed. The expectation of a certain summand $X_{I_{k+1}} / p_{I_{k+1}}$ will be

$$
\mathbb{E} \frac{X_{I_{k+1}}}{p_{I_{k+1}}} = \mathbb{E}\left[\mathbb{E}\left[\frac{X_{I_{k+1}}}{p_{I_{k+1}}^{(k)}} \frac{p_{I_{k+1}}^{(k)}}{p_{I_{k+1}}} \mid \mathcal{F}_k\right]\right] = \mathbb{E}\left[\frac{1}{\sum_{i \in V - S_k} p_i} \sum_{i \in V - S_k} X_i\right], \tag{5}
$$

where $\mathcal{F}_k$ is the $\sigma$-algebra generated by the random indices inside the corresponding set $S_k$, $\forall k = 0, 1, \cdots, m - 1$, and the second equation holds because $\frac{p_{I_{k+1}}^{(k)}}{p_{I_{k+1}}} = \frac{1}{\sum_{i \in V - S_k} p_i}$ is $\mathcal{F}_k$-measurable. The expectation is in general unequal to the target $\sum_{i=1}^n X_i$ for $k > 0$, except for some extreme conditions such as all-equal $p_i$'s. The bias in each summand (except for the first term with $k = 0$) accumulates and results in the ultimate bias in the given estimation.

## 5.2 A REMEDY TO ATTAIN UNBIASED ESTIMATION

The bias induced by the sequential WRS algorithm has been extensively analyzed by many other studies, especially the ones on stochastic gradient estimators (Liang et al., 2018; Liu et al., 2019; Kool et al., 2019). Given a sequence of random indices sampled through WRS, as far as we know there are two genres to assign coefficients to summands in Equation (4). Both of the two methods relate to the *stochastic sum-and-sample* estimator (Liang et al., 2018; Liu et al., 2019), which can be derived from Equation (5). Using the fact $\mathbb{E} \frac{\boldsymbol{X}_{I_{k+1}}}{p_{I_{k+1}}} \sum_{i \in V - S_k} p_i = \mathbb{E} \left[ \sum_{i \in V - S_k} \boldsymbol{X}_i \right]$, a stochastic sum-and-sample estimator of $\sum_{i=1}^n \boldsymbol{X}_i$ can be immediately constructed as

$$\boldsymbol{\Pi}_{k+1} = \sum_{j \in S_k} \boldsymbol{X}_i + \frac{\boldsymbol{X}_{I_{k+1}}}{p_{I_{k+1}}^{(k)}}, \forall k = 0, 1, \cdots m - 1. \tag{6}$$

(The proof of unbiasedness is brief and provided by Kool et al. (2019, Appendix C.1).) To minimize the variance, Liang et al. (2018); Liu et al. (2019) develop the first genre to focus on the last estimator $\boldsymbol{\Pi}_m$ and propose methods to pick the initial $m - 1$ random indices. Kool et al. (2019, Theorem 4) turn to the second genre which utilize Rao-Blackwellization (Casella & Robert, 1996) of $\boldsymbol{\Pi}_m$.

In fast training for GCN, both of the two genres are somewhat inefficient from a practitioner's perspective. The first genre works well when $\sum_{i \in S_{m-1}} p_i$ is close to 1, otherwise the last term in $\boldsymbol{\Pi}_m$, $\frac{\boldsymbol{X}_{I_{k+1}}}{p_{I_{k+1}}^{(k)}}$, will brings in large variance and reduce the sample efficiency; for the second genre, the time cost to perform Rao-Blackwellization (Kool et al., 2019) is extremely high ($\mathcal{O}(2^m)$ even with approximation by numerical integration) and conflicts with the purpose of fast training. To overcome the issues of the two existing genres, we propose an iterative method to fully utilize each estimator $\boldsymbol{\Pi}_{k+1}$ with acceptable runtime to decide the coefficients for each term in Equation (4).

Conceptually, we recursively perform the weighted averaging below:

$$\boldsymbol{Z}_0 := 0, \quad \boldsymbol{Z}_{k+1} := (1 - \alpha_{k+1})\boldsymbol{Z}_k + \alpha_{k+1}\boldsymbol{\Pi}_{k+1}, \forall k = 0, 1, \cdots, m - 1,$$

where $\alpha_{k+1}$ is a constant decided by $k$. When $\alpha_1 = 1$, $\boldsymbol{Z}_1 = \boldsymbol{\Pi}_1 = \frac{\boldsymbol{X}_{I_1}}{p_{I_1}}$ is unbiased and the unbiasedness of $\boldsymbol{Z}_m$ can be obtained by induction as each $\boldsymbol{\Pi}_{k+1}$ is unbiased as well. For the choice of $\alpha_{k+1}$'s, we intentionally specify $\alpha_{k+1} = \frac{n}{(n-k)(k+1)}$, motivated by the preference that if all $p_i$'s are $1/n$, the output coefficients of the algorithm will be all $1/m$, the same as the ones in an SRS setting.

In practice, it suffices for the debiasing algorithm to find the coefficients for each $\boldsymbol{X}_{k+1}$, which can be obtained in advance of the final computation of $\boldsymbol{Z}_m$. We, therefore, adapt the preceding recursive estimation to iterative updates of coefficients, and conclude the procedure in Algorithm 1 in Appendix A. The time complexity of Algorithm 1 is $\mathcal{O}(m^2)$, as in iteration $k + 1$ we update the coefficients for the first $k + 1$ random indices sampled. The time complexity is comparable to the one of embedding aggregation in layer-wise sampling, as shown in Appendix D.

## 5.3 EFFECTS OF DEBIASING

As an immediate result, our proposed debiasing method can significantly improve the matrix approximation error in a single step of embedding aggregation. In Figure 1, we observe in all the five datasets, the debiased variant of LADIES has a smaller error, and its error ordinarily drops with the number of sub-samples. When combining the debiasing with the new probabilities raised in Section 2.2, the effect of debiasing is significant only when the graphs (ogbn-arxiv and ogbn-mag) are sparse and when the sub-sample sizes are large enough. We have the following conjectures for this phenomenon. The insignificant improvement is caused by the flat distribution of our proposed probabilities, as the debiasing procedure will maintain the estimation if the sampling probabilities are all-equal. For sparse graphs, however, the number of sampled columns ($m$) is comparable to the neighborhood size of the batch nodes ($n$), and as the biases accumulate the effect of the debiasing algorithm emerges.

We further comment that debiasing is not a panacea for GCN accuracy, and the results in Figure 1 cannot fully predict the corresponding GCN accuracy. Conceptually, the columns sampled are

Table 1: Summary of prediction accuracy on different datasets. The accuracy is reported in the form of a percentage (%).

| | Reddit | ogbn-arxiv | ogbn-proteins | ogbn-products | ogbn-mag |
|---|---|---|---|---|---|
| Full-batch | 93.81±0.18 | 66.39±0.25 | 65.71±0.11 | 68.33±0.16 | 29.60±0.27 |
| Node-wise (2) | 92.13±0.27 | 64.51±0.30 | 65.76±0.18 | 68.71±0.07 | 29.05±0.45 |
| Node-wise (10) | 94.41±0.07 | 66.47±0.19 | 66.34±0.10 | 69.57±0.20 | 29.54±0.27 |
| VR-GCN (2) | 94.62±0.04 | 67.49±0.25 | 67.45±0.02 | 70.90±0.28 | 28.99±0.40 |
| VR-GCN (10) | 94.36±0.05 | 66.50±0.33 | 66.39±0.06 | 69.81±0.31 | 29.66±0.68 |
| GraphSAINT | 89.47±0.83 | 60.58±0.62 | 66.33±0.07 | 62.77±1.04 | 24.77±0.88 |
| FastGCN | 44.46±2.30 | 25.44±0.82 | 52.44±1.88 | 26.98±0.42 | 7.13±0.48 |
| FastGCN (2) | 60.31±0.70 | 30.23±1.10 | 58.80±1.06 | 31.58±0.70 | 5.85±0.57 |
| LADIES | 73.86±0.17 | 60.95±0.31 | 68.28±0.05 | 52.97±1.11 | 24.79±0.48 |
| w/ flat | 90.04±0.11 | 62.76±0.26 | 68.26±0.06 | 62.64±0.10 | 27.30±0.27 |
| w/ debiased | 86.73±0.36 | 61.55±0.40 | 68.87±0.09 | 55.92±0.92 | 25.74±0.80 |
| w/ flat & debiased | 89.34±0.40 | 61.90±0.43 | 67.64±0.15 | 62.57±0.22 | 27.41±0.28 |
| LADIES (2) | 88.34±0.11 | 64.01±0.39 | 68.17±0.10 | 65.24±0.40 | 28.59±0.39 |
| w/ flat | 93.64±0.19 | 66.56±1.84 | 68.10±0.07 | 68.47±0.25 | 29.58±0.19 |
| w/ debiased | 92.75±0.22 | 65.93±0.27 | 69.14±0.15 | 67.18±0.24 | 30.08±0.28 |
| w/ flat & debiased | 93.59±0.09 | 66.22±0.10 | 67.75±0.11 | 68.49±0.06 | 29.88±0.34 |

mainly decided by the sampling probabilities, and a debiasing procedure solely adjusts the weights for different columns. This adjustment partially overlaps with the updates of the parameters $W^{(l)}$'s in a GCN, and therefore the improvement of debiasing to the GCN accuracy will be limited. We claim debiasing should be considered as a means to save some parameter updates, more than a technique designed to improve the GCN accuracy. We will empirically verify the effect of debiasing in the next section.

## 6 EXPERIMENTS

In this section, we empirically evaluate the performance of each method on five node prediction datasets: Reddit, ogbn-arxiv, ogbn-proteins, ogbn-mag, ogbn-products (See Table 2). We denote "LADIES+flat", "LADIES+debiased", and "LADIES+flat+debiased" respectively as the variants of LADIES with the improvements from Section 4, from Section 5, and from both. We compare our methods to the original GCN with mini-batch stochastic training (denoted by full-batch), two layer-wise sampling methods: FastGCN and LADIES. Apart from that, we also implement the other common fast GCN training methods, including GraphSAGE (Hamilton et al., 2017) (vanilla node-wise sampling while keep using the GCN architecture), VR-GCN (Chen et al., 2018a), and subgraph sampling method GraphSAINT (Zeng et al., 2019).

For the training settings, we use a 2-layer GCN for each task with an ADAM optimizer of learning rate 0.001. (Due to limited computational resources, we have to use the shallow GCN since the full-batch method and node-wise sampling methods require much more GPU memory even when $L = 3$.) The number of hidden variables is 256 and the batch size is 512. For layer-wise sampling methods, we adopt both the classical setting that in each layer the numbers of columns sampled are equal to the batch size 512, and an "increasing" setting (denoted with (2)) that twice nodes will be sampled in the next layer. For the node-wise sampling methods, we combine the settings in the original papers of GraphSAGE (Hamilton et al., 2017) VR-GCN (Chen et al., 2018a) that we separately set the number of neighbors per node as 2 (denoted with (2)) and 10 (denoted with (10)). For the subgraph sampling method GraphSAINT (Zeng et al., 2019), the subgraph size is by default equal to the batch size. The experiment results are reported in Table 1, with the means and associated standard deviations based on 5 runs. More details of the settings are deferred to Appendix C.1.

### 6.1 TRAJECTORIES OF MODEL CONVERGENCE

We first compare the convergence rates of different methods. The results on Reddit, ogbn-proteins, and ogbn-products are shown in Figure 3, and the results on other datasets are deferred to Figure 4 in Appendix C.2 due to limited space.

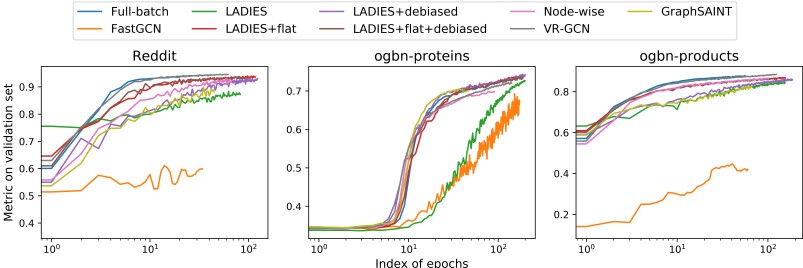

Figure 3: Metrics in each epoch. The layer-wise sampling methods follow the "increasing" setting (denoted with (2) in Table 1); for the node-wise sampling methods, the number of neighbors is 2.

In Figure 3 (and Figure 4), we observe LADIES (the green curve) tends to move fast at the beginning of training while slow down after the first stage. In contrast, the debiased variants of LADIES have convergence patterns similar to the outstanding while time-consuming methods Full-batch and VR-GCN, and seem to accelerate the training of LADIES at the local convergence stage. Figure 3 also demonstrates that in most cases, "LADIES+debiased" converges slower than "LADIES+flat", which indicates that regarding convergence, the effect of debiasing is not as important as choosing a proper sampling scheme. This observation matches our analysis in Section 5.3 that debiasing can save some turns of parameter updates and sampling probabilities decide the overall quality of the approximation to embedding aggregations.

## 6.2 PREDICTION ACCURACY

The prediction accuracy on test sets of different datasets is reported in Table 1. We observe the new sampling probabilities and the debiasing algorithm show consistent improvement over existing layer-wise sampling methods, FastGCN and LADIES. In addition, the experiments further verify the previous observation that the two remedies we apply have the overlapping effect, which is implied in Figure 1 and Figure 3 that (when sub-sample size is small) the improvement of "LADIES+flat+debiased" over its ablation method "LADIES+flat" is insignificant. As for the four benchmark methods, the full-batch method and two node-wise sampling methods have outstanding performance on the accuracy, as they benefit from the vast amount of computation. The graph-SAINT method demonstrates a tendency to perform well on the dataset with a high average degree, such as ogbn-proteins. We comment that when the amount of computation is controlled, layer-wise sampling can enjoy similar prediction accuracy to node-wise sampling, as shown in Table 1.

We close this section with a remark on the seemingly strange phenomenon that some efficient GCNs aim to mimic the original full-batch GCN while having a higher prediction accuracy on some datasets. We speculate the reason behind this phenomenon is that a good approximation can recover the principal components in the original embedding matrix, and also restrain the noise via the sparse / low-rank structure. A similar observation (Sanyal et al., 2018) is found in Convolutional Neural Networks (CNN) as well, that applying a low-rank regularizer, such as SVD, to the representation of the intermediate layers can improve the prediction errors of the CNN models.

## 7 CONCLUSION

We carefully analyze layer-wise sampling in this work and make two improvements to current layer-wise sampling models. We first show that a conservative choice of sampling probabilities outperforms the existing ones. The latter probabilities assume the $L^2$ norm of embedding is proportional to the norm of the corresponding column of the renormalized Laplacian matrix, which is usually too strong in practice. We further propose a new correction method to debias the estimation in layer-wise sampling through iterative updates of coefficients for columns sampled. The empirical experiments justify our proposed method and show that our method achieves high accuracy close to the SOTA node-wise sampling method, VR-GCN.

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

## A    ALGORITHM TO SPECIFY THE COEFFICIENTS FOR DEBIASING

The algorithm described in Section 5.2 is provided in this section.

---

**Algorithm 1:** Iterative updates of coefficients to construct the ultimate debiased estimator $Z_m$

---

**Input:** probabilities $\{p_i\}_{i=1}^n$, random indices $\{I_{k+1}\}_{k=0}^{m-1}$ generated by WRS with $\{p_i\}_{i=1}^n$
**Output:** a length $m$ coefficient vector $\boldsymbol{\beta}$
Initialize the vector $\boldsymbol{\beta} = \mathbf{0} \in \mathbb{R}^m$, sum of probabilities $p_S = 0$;
**for** $k \leftarrow 0$ **to** $m-1$ **do**
    $\alpha = \frac{n}{(n-k)(k+1)}$;
    $\boldsymbol{\beta}[k] = \alpha(1 - p_S)/p_{I_{k+1}}$;
    **for** $j \leftarrow 0$ **to** $k-1$ **do**
        $\boldsymbol{\beta}[j] = (1 - \alpha)\boldsymbol{\beta}[j] + \alpha$;
    **end**
    $p_S = p_S + p_{I_{k+1}}$;
**end**
return $\boldsymbol{\beta}$;

---

Table 2: Summary of datasets. Each undirected edge is counted once. Each node in ogbn-protein has 112 binary labels. "D." refers to the average degree of the graph. "Feat" refers to the number of features. "Split Ratio" refers to the ratio of training/validation/test data.

| Dataset | Nodes | Edges | D. | Feat. | Classes | Tasks | Split Ratio | Metric |
|---|---|---|---|---|---|---|---|---|
| Reddit | 232,965 | 11,606,919 | 50 | 602 | 41 | 1 | 66/10/24 | F1-score |
| ogbn-arxiv | 160,343 | 1,166,243 | 13.7 | 128 | 40 | 1 | 54/18/28 | Accuracy |
| ogbn-proteins | 132,534 | 39,561,252 | 597.0 | 8 | binary | 112 | 65/16/19 | ROC-AUC |
| ognb-mag | 736,389 | 5,396,336 | 7.3 | 128 | 349 | 1 | 85/9/6 | Accuracy |
| ogbn-products | 2,449,029 | 61,859,140 | 50.5 | 100 | 47 | 1 | 8/2/90 | Accuracy |

## B    INFORMATION OF DATASETS

Table 2 summaries the information of datasets used in our experiments.

## C    ADDITIONAL EXPERIMENT DETAILS

### C.1    ADDITIONAL DETAILS ON EXPERIMENTAL SETUPS

In this section, we describe the additional details of experiment setups for Section 6. All the models are implemented by Pytorch. We use one Tesla V100 SXM2 32GB GPU with 10 CPU threads to train the models in Section 6. Our implementation of Full-batch method, FastGCN, LADIES are adapted from the codes by Zou et al. (2019); our implementation of vanilla node-wise sampling, VR-GCN, GraphSAINT is adapted from the codes by Cong et al. (2020). For the vanilla node-wise sampling method, there are several variants of structures Ying et al. (2018) while we fix the model structure as GCN in our experiments for fair comparison. We use ELU as the activation function in the convolutional layer for all the models: $ELU(x) = x$ for $x > 0$, $ELU(x) = \exp(x) - 1$ for $x \leq 0$. We choose dropout rate as 0.2, which means 20 percents of units are randomly dropped during the training. Validation and testing are performed with Full-batch inference (using all possible neighbors) on validation and testing nodes. Note that some existing Pytorch implementations of GCNs involve several ad-hoc tricks, such as row-normalizing sampled Laplacian matrix. For the accuracy evaluation experiments in Section 6, we stop training when the validation F1 score does not increase for 200 batches. For a fair comparison, we remove certain tricks in our experiments, such as normalization of each rows in the sampled Laplacian matrix in layer-wise sampling. Such a trick may help in the practice but it might not be compatible to some other methods and is out of the focus of our study. We use the metrics in Table 2 to evaluate the accuracy of each method. Concretely, Reddit is a multi-class classification task and we use the Micro-F1 score with function "sklearn.metrics.f1_score". For OGB data, we use the built-in evaluator function in module `ogb` by Hu et al. (2020).

### C.2    ADDITIONAL RESULTS ON MODEL CONVERGENCE

The convergence results on ogbn-arxiv and ogbn-mag are provided in Figure 4. The setting of each model is the same as in Figure 3.

### C.3    ADDITIONAL REGRESSION EXPERIMENTS

For the regression experiments in Section 4.1, we train a 3-layer GCN without sampling, i.e., full-batch SGD training, with 256 hidden variables per layer. The batch size is 512. Early stopping training policy is applied. We also remark that these experiments are conducted to check the assumption of importance sampling, rather than pursuing SOTA performance. When we finish training the model, the norms of rows in $\boldsymbol{HW}$ are extracted through a full-batch inference with all training nodes. Hence, we only record the pairs $(x_i^{(l)}, y_i)$'s for sampled nodes in regression, where $i$ indicates the index of the sampled nodes in the $l$-th layer.

Figure 5 is a supplementary to Figure 2. The assumption: $\|(HW)_{[i]}\| \propto \|P^{[i]}\|$ does not hold on all of these datasets. We do not have the regression result on the ogbn-product dataset since the training of 3-layer GCN fails due to the memory limitation.

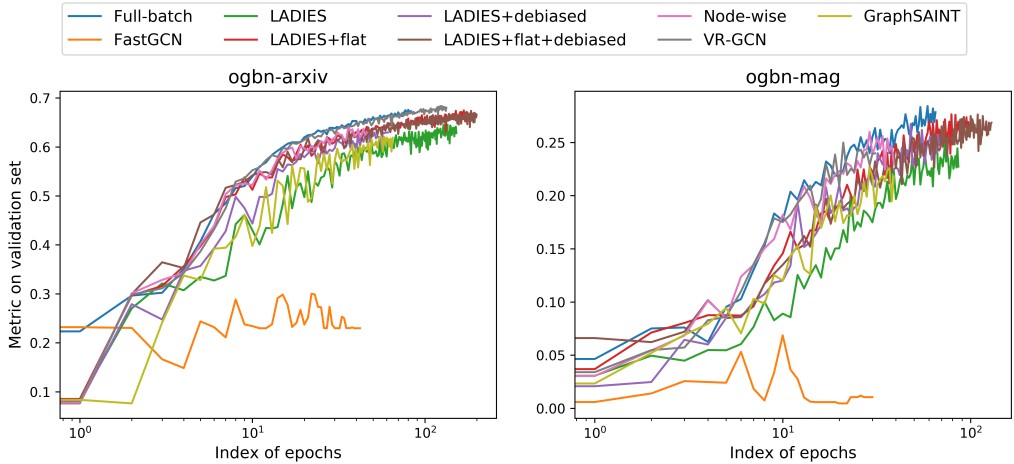

Figure 4: Metrics in each epoch.

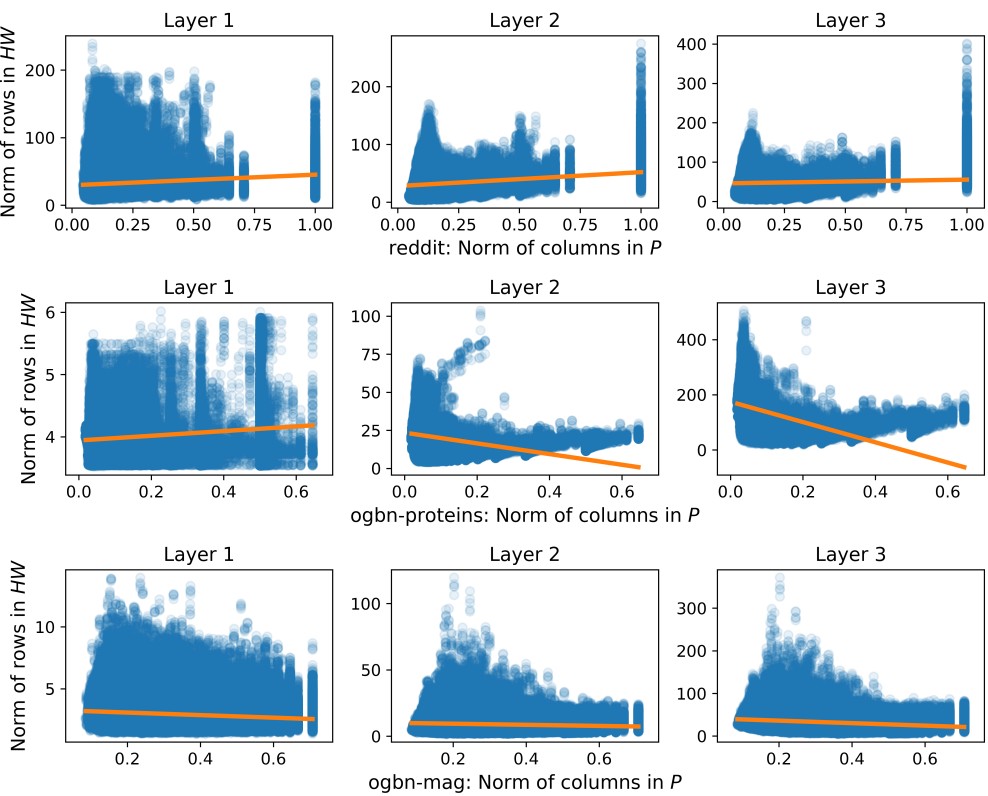

Figure 5: Regression on Reddit, ogbn-protein, ogbn-mag datasets. GCN is trained with full-batch SGD.

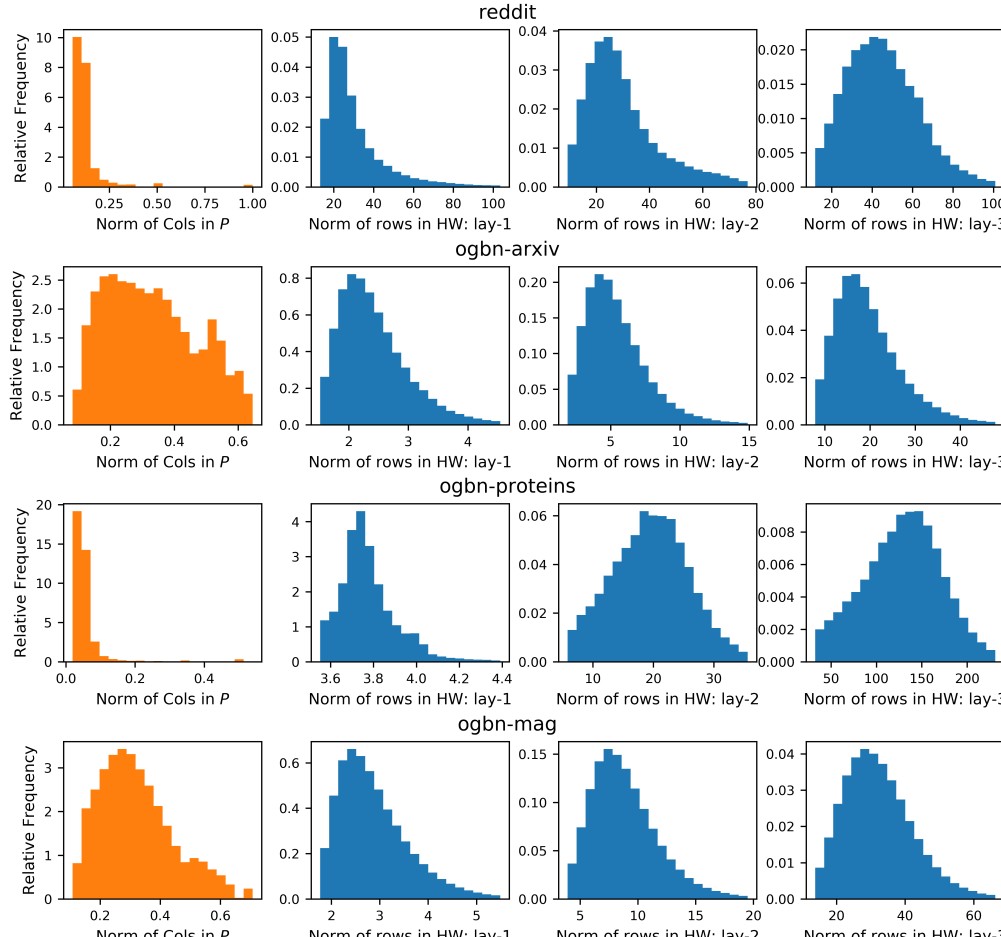

Figure 6: Distribution of $\|P^{[i]}\|$ and $\|(HW)_{[i]}\|$ for Reddit, ogbn-arxiv, ogbn-protein and ogbn-mag.

Figure 6 demonstrates the distribution of $\|\boldsymbol{P}^{[i]}\|$ and $\|(\boldsymbol{HW})_{[i]}\|$ (layer 1, 2, 3) for Reddit, ogbn-arxiv, ogbn-protein and ogbn-mag datasets. The $\|(\boldsymbol{HW})_{[i]}\|$'s are obtained from the experiment in Section C.3. The outliers larger than the 99.9% quantile or small than the 0.1% quantile are removed.

As shown in the histograms, our analysis regarding Assumption 1 tends to hold generally on these datasets. For the norms of columns in $\boldsymbol{P}$ (as a replacement for $\boldsymbol{QP}$ for clarity), we observe there are some columns with large norms far beyond the average. Those columns contribute a lot to the quadratic mean, which results in a huge $C_1$ in Assumption 1. In contrast, the norms of rows in $\boldsymbol{HW}$ concentrate around their average, inducing a small $C_2$. Those facts together with Assumption 1 and Lemma 1 in the main paper explain why our proposed sampling probabilities are more proper for some real datasets.

## D    TIME COMPLEXITY ANALYSIS

We analyze the complexity of vanilla node-wise sampling and layer-wise sampling in this section. The analysis is adapted from the work by Zou et al. (2019), but we show a lighter bound for layer-wise sampling. For $l$ such that $0 \le l \le L - 1$, the propagation formulas for sampling based GCN can be formulated as:

$$\tilde{\boldsymbol{Z}}^{(l+1)} = \bar{\boldsymbol{P}}^{(l)} \tilde{\boldsymbol{H}}^{(l)} \boldsymbol{W}^{(l)},$$

where $\tilde{\boldsymbol{H}}^{(l)} = \tilde{\boldsymbol{Z}}^{(l)} \in \mathbb{R}^{s_l \times p}$, $\hat{\boldsymbol{P}}^{(l)} \in \mathbb{R}^{s_{l+1} \times s_l}$, $W^{(l)} \in \mathbb{R}^{p \times p}$. In particular, for LADIES, $\bar{\boldsymbol{P}}^{(l)}_{LADIES} = \boldsymbol{Q}^{(l+1)} \boldsymbol{P} \boldsymbol{S}^{(l)}$.

Table 3: The time complexity for computation for $L$-layer GCN training by layer-wise sampling and node-wise sampling . The first column refers to the matrix operation type, nodes aggregation or linear transformation.

|  | Layer-wise | Node-wise |
| --- | --- | --- |
| Nodes Aggregation | $\mathcal{O}(scpL)$ | $\mathcal{O}(sb^L p)$ |
| Linear Transformation | $\mathcal{O}(sp^2 L)$ | $\mathcal{O}(sb^{L-1} p^2)$ |

For simplicity, we suppose that the number of hidden variables in each layer is fixed as $p$, the same as the dimension of $\boldsymbol{H}^{(0)}$. The batch size and the numbers of nodes sampled in each layer equal are set all equal to a fixed constant $s$. We assume the number of sampled neighbors per node in node-wise sampling is $b$. We denote the maximal degree of all the nodes in the graph as $c$. The computational cost of the propagation comes from two parts: the linear transformation, a dense matrix product, $\tilde{\boldsymbol{H}}^{(l)} \boldsymbol{W}^{(l)}$ and the node aggregation, a sparse matrix product, $\bar{\boldsymbol{P}}^{(l)} (\tilde{\boldsymbol{H}}^{(l)} \boldsymbol{W}^{(l)})$. The time complexity is summarized in Table 3. We additionally comment although the time cost of two parts both linearly depend on the number of nodes involved (number of non-zero elements in $\boldsymbol{Q}^{(l+1)}$), the node aggregation part usually dominates since the sparse matrix product involved is less efficient than the dense matrix product involved in a modern computer.

The linear transformation, $\tilde{\boldsymbol{H}}^{(l)} \boldsymbol{W}^{(l)}$ is dense matrix production. The cost depends on the shape of two matrices, and is given as $\mathcal{O}(s_l p^2)$. LADIES fixes $s_l$ as $s$ for each layer, so $\mathcal{O}(s_l p^2) = \mathcal{O}(sp^2)$. For node-wise sampling $s_l = sb^{L-l}$, since number of node exponentially grows. Thus by summation over all the layers, we have the results in the second row in Table 3.

The node aggregation, $\bar{\boldsymbol{P}}^{(l)} (\tilde{\boldsymbol{H}}^{(l)} \boldsymbol{W}^{(l)})$ is a sparse matrix production, since $\bar{\boldsymbol{P}}^{(l)}$ is sparse. For simplicity, we denote $\tilde{\boldsymbol{H}}^{(l)} \boldsymbol{W}^{(l)}$ as $\boldsymbol{C}^{(l)} \in \mathbb{R}^{s_l \times p}$. Thus the time complexity of this sparse matrix production becomes $\mathcal{O}(nnz^{(P_l)} p)$, where $nnz^{(P_l)}$ is the number of non-zero entries in $\bar{\boldsymbol{P}}^{(l)}$. For layer-wise sampling, since we sample $s$ nodes for each layer and each node has at most $c$ neighbors, so $nnz^{(P_l)} \leq sc$. For node-wise sampling, since each node has $b$ neighbors and the neighbors are not shared by all the nodes in each layer, $nnz^{(P_l)} = bs_l = sb^{L+1-l}$. By summation over all the layers, we attain the results in the first row of Table 3.

## E  RESULTS IN APPROXIMATE MATRIX MULTIPLICATION

In this section, we revisit approximate matrix multiplication to derive the previous layer-wise sampling methods. Specifically, the sampling matrix $\boldsymbol{S}$ used in FastGCN and LADIES can be decomposed as $\boldsymbol{S} = \boldsymbol{\Pi}\boldsymbol{\Pi}^T$, where $\boldsymbol{\Pi} \in \mathbb{R}^{n \times d}$ is a sub-sampling sketching matrix defined as follows:

**Definition E.1** (Sub-sampling sketching matrix). *Consider a discrete distribution which draws $i$ with probability $p_i > 0, \forall i \in [n]$. For a random matrix $\boldsymbol{\Pi} \in \mathbb{R}^{n \times d}$, if $\boldsymbol{\Pi}$ has i.i.d. columns and each column $\boldsymbol{\Pi}^{(j)}$ can randomly be $\frac{1}{\sqrt{dp_i}} \boldsymbol{e}_i$ with probability $p_i$, where $\boldsymbol{e}_i$ is the $i$-th column of the $n$-by-$n$ identity matrix $\boldsymbol{I}_n$, then $\boldsymbol{\Pi}$ is called a sub-sampling sketching matrix with sub-sampling probabilities $\{p_i\}_{i=1}^n$.*

With this definition, we introduce a result in AMM to construct the sub-sampling sketching matrix, which coincides with the conclusion in FastGCN and LADIES.

**Theorem E.1** (Theorem 1 (Drineas et al., 2006)). *Suppose $\boldsymbol{B} \in \mathbb{R}^{n_B \times n}$, $\boldsymbol{C} \in \mathbb{R}^{n \times n_C}$, the number of sub-sampled columns $d \in \mathbb{Z}^+$ such that $1 \leq d \leq n$, and the sub-sampling probabilities $\{p_i\}_{i=1}^n$ are such that $\sum_{i=1}^n p_i = 1$ and such that for a quality coefficient $\beta \in (0, 1]$*

$$p_i \geq \beta \frac{\|\boldsymbol{B}^{[i]}\| \|\boldsymbol{C}_{[i]}\|}{\sum_{i'=1}^n \|\boldsymbol{B}^{[i']}\| \|\boldsymbol{C}_{[i']}\|}, \forall i \in [n]. \tag{7}$$

*Construct a sub-sampling sketching matrix $\boldsymbol{\Pi} \in \mathbb{R}^{n \times d}$ with sub-sampling probabilities $\{p_i\}_{i=1}^n$ as in Definition E.1, and let $\boldsymbol{B}\boldsymbol{\Pi}\boldsymbol{\Pi}^T \boldsymbol{C}$ be an approximation to $\boldsymbol{B}\boldsymbol{C}$. Let $\delta \in (0, 1)$ and $\eta =$*

$1 + \sqrt{(8/\beta) \log(1/\delta)}$. *Then with probability at least* $1 - \delta$,

$$\|\boldsymbol{BC} - \boldsymbol{B\Pi\Pi}^T\boldsymbol{C}\|_F^2 \leq \frac{\eta^2}{\beta d}\|\boldsymbol{B}\|_F^2\|\boldsymbol{C}\|_F^2. \tag{8}$$

**Remark.** The theorem is closely related to Lemma 1 in Appendix B of LADIES, which studies the variance $\mathbb{E}\|\boldsymbol{BC} - \boldsymbol{BSC}\|_F^2$. For the choice of sub-sampling probabilities, Equation (7) reproduces the conclusion in FastGCN and LADIES, when we respectively take $\boldsymbol{B}$ as $\boldsymbol{P}$ and $\boldsymbol{QP}$.

## F  PROOF OF LEMMA 1 AND SOME REMARKS

To prove Lemma 1 in the main paper, we first adapt a technical lemma (Zou et al., 2019, Lemma 1), which relates the sampling matrix to the variance (expectation of squared Frobenius norm) of the approximate matrix multiplication.

**Lemma F.1** (Adapted from Lemma 1 (Zou et al., 2019)). *Given two matrices* $\boldsymbol{B} \in \mathbb{R}^{n_B \times n}$ *and* $\boldsymbol{C} \in \mathbb{R}^{n \times n_C}$, *for any* $i \in [n]$ *define the positive probabilities* $p_i$'s *such that* $\sum_{i=1}^{n} p_i = 1$. *We further require the probability* $p_i = 0$ *if and only if the corresponding column* $\boldsymbol{B}^{[i]}$ *or row* $\boldsymbol{C}_{[i]}$ *is all-zero. The sub-sampling sketching matrix* $\boldsymbol{\Pi} \in \mathbb{R}^{n \times d}$ *is generated accordingly. Let* $\boldsymbol{S} := \boldsymbol{\Pi\Pi}^T$, *it holds that*

$$\mathbb{E}_{\boldsymbol{S}}\left[\|\boldsymbol{BSC} - \boldsymbol{BC}\|_F^2\right] = \frac{1}{d}\left(\sum_{i:p_i>0}\frac{1}{p_i}\left\|\boldsymbol{B}^{[i]}\right\|^2 \cdot \left\|\boldsymbol{C}_{[i]}\right\|^2 - \|\boldsymbol{BC}\|_F^2\right)$$

*where* $d$ *is the number of samples.*

With the lemma above, the proof of Lemma 1 in the main paper is provided as follows.

*Proof.* Recall the notation in Lemma 1 in the main paper is simplified as $\boldsymbol{B} := \boldsymbol{QP}, \boldsymbol{C} := \boldsymbol{HW}$. As the union of neighbors of nodes in $\boldsymbol{Q}$ cannot cover all the nodes, some columns in $\boldsymbol{B}$ are all-zero, and we accordingly define an $\boldsymbol{Q}$-measurable matrix $\boldsymbol{L}$ as in Lemma F.1. We have

$$\mathbb{E}\left[\|\boldsymbol{BS_1C} - \boldsymbol{BC}\|_F^2\right] = \mathbb{E}_{\boldsymbol{Q}}\left[\mathbb{E}_{\boldsymbol{S_1}}\left(\|\boldsymbol{BS_1C} - \boldsymbol{BC}\|_F^2|\boldsymbol{Q}\right)\right]$$

$$= \frac{1}{d}\mathbb{E}_{\boldsymbol{Q}}\left[\sum_{i:p_i>0}\frac{1}{p_i}\left\|\boldsymbol{B}^{[i]}\right\|^2 \cdot \left\|\boldsymbol{C}_{[i]}\right\|^2 - \|\boldsymbol{BC}\|_F^2\right].$$

where the second equation holds as we apply Lemma F.1 to the inner expectation in the right-hand side of the first line. Plugging $p_i \propto \|\boldsymbol{B}^{[i]}\|$ (Equation (3) in the main paper) into the preceding probabilities $p_i$'s, we reach

$$\mathbb{E}\left[\|\boldsymbol{BS_1C} - \boldsymbol{BC}\|_F^2\right] = \frac{\mathbb{E}_{\boldsymbol{Q}}\left[\left(\sum_{i:p_i>0}\left\|\boldsymbol{B}^{[i]}\right\|\right)\left(\sum_{i:p_i>0}\left\|\boldsymbol{B}^{[i]}\right\|\left\|\boldsymbol{C}_{[i]}\right\|^2\right)\right]}{d} - \frac{\mathbb{E}_{\boldsymbol{Q}}\left[\|\boldsymbol{BC}\|_F^2\right]}{d}$$

$$= \frac{1}{d}\mathbb{E}_{\boldsymbol{Q}}\left[\left(\sum_{i=1}^{n}\left\|\boldsymbol{B}^{[i]}\right\|\right)\left(\sum_{i=1}^{n}\left\|\boldsymbol{B}^{[i]}\right\|\left\|\boldsymbol{C}_{[i]}\right\|^2\right)\right] - \frac{1}{d}\mathbb{E}_{\boldsymbol{Q}}\left[\|\boldsymbol{BC}\|_F^2\right].$$

As computed by Zou et al. (2019), the variance of LADIES is similarly given as

$$\mathbb{E}\left[\|\boldsymbol{BS_0C} - \boldsymbol{BC}\|_F^2\right] = \frac{1}{d}\mathbb{E}_{\boldsymbol{Q}}\left[\left(\sum_{i:p_i>0}\left\|\boldsymbol{B}^{[i]}\right\|^2\right)\left(\sum_{i:p_i>0}\left\|\boldsymbol{C}_{[i]}\right\|^2\right)\right] - \frac{1}{d}\mathbb{E}_{\boldsymbol{Q}}\left[\|\boldsymbol{BC}\|_F^2\right].$$

Consequently, to prove the lemma it suffices to show that

$$\left(\sum_{i:p_i>0}\left\|\boldsymbol{B}^{[i]}\right\|\right)\left(\sum_{i:p_i>0}\left\|\boldsymbol{B}^{[i]}\right\|\left\|\boldsymbol{C}_{[i]}\right\|^2\right) \leq \left(\sum_{i:p_i>0}\left\|\boldsymbol{B}^{[i]}\right\|^2\right)\left(\sum_{i:p_i>0}\left\|\boldsymbol{C}_{[i]}\right\|^2\right), \tag{9}$$

and the inequality above follows with Assumption 1. Specifically, plugging the inequality $\|\boldsymbol{C}_{[i]}\|^2 \le C_2\|\boldsymbol{C}\|_F^2/n, \forall i \in [n]$ in the left-hand-side above, we have

$$\left(\sum_{i:p_i>0}\left\|\boldsymbol{B}^{[i]}\right\|\right)\left(\sum_{i:p_i>0}\left\|\boldsymbol{B}^{[i]}\right\|\left\|\boldsymbol{C}_{[i]}\right\|^2\right) \le \left(\sum_{i=1}^{n}\left\|\boldsymbol{B}^{[i]}\right\|\right)^2 \frac{C_2}{n}\|\boldsymbol{C}\|_F^2 = \frac{m}{C_1}\|\boldsymbol{B}\|_F^2 \frac{C_2}{nm}m\|\boldsymbol{C}\|_F^2,$$

in which the last equation comes from the definition $C_1 := \frac{\|\boldsymbol{B}\|_F^2/m}{\left(\sum_{i=1}^{n}\|\boldsymbol{B}^{[i]}\|/m\right)^2}$. To close the proof, we utilize the inequality $\frac{1}{C_2}\|\boldsymbol{C}\|_F^2/n \le \|\boldsymbol{C}_{[i]}\|^2$ and bound $m\|\boldsymbol{C}\|_F^2$ by $nC_2\sum_{i:p_i>0}\left\|\boldsymbol{C}_{[i]}\right\|^2$. Finally we attain Equation (9) with the core assumption $\frac{C_1}{C_2^2} \ge 1$. $\diamond$

**Remark.** In Assumption 1 we indeed implicitly assume $\|\boldsymbol{B}^{[i]}\|$'s follow a long-tail distribution that most norms are around the average while a few columns have large norms. The high non-uniformity makes the average of squared norms much larger than the square of averaged norms. For $\|\boldsymbol{C}_{[i]}\|$'s, considering the normalization techniques (such as batch or layer normalization) to stabilize the scale of the parameters, they tend to not vary widely, which implies a small $C_2$. The numerical experiments on the comparison of approximation error (see Figure 1) and the histograms of the norms in trained models shown in Figure 6 further validate the assumption. Based on the empirical analysis above, we claim the assumption is mild and tends to hold at least for some datasets.

## G SUPPLEMENTARY EXPERIMENTS OF REBUTTAL REVISION

### G.1 SUPPLEMENTARY REGRESSION EXPERIMENTS

In addition to the regression experiments in Section 4.1 and Appendix C.3, we also present the regression results in Figure 7 and Figure 5, for a 3-layer GCN with our LADIES+flat+debias sampler and LADIES sampler respectively Both of them have 512 nodes sampled at each layer and keep other hyper-parameters the same as regression experiments. The regression results in the following figures show similar patterns to Figures 2 and 5.

### G.2 SAMPLING TIME AND TRAINING TIME

We compare the sampling time per batch for 1-layer GCN with layer-wise sampling methods (Fast-GCN, LADIES, and our proposed methods) and GraphSAGE. The time is presented in milliseconds. The batch size is 512, and the number of sampled nodes is 512 or 1024. The average sampling time (followed by standard deviation) over 200 batches is presented in Table 4. We note that the sampling time may involve some overhead costs. For example, the input Laplacian matrix is Scipy-spare-matrix on the CPU, while in sampling, it is converted to a PyTorch-sparse-matrix.

By Table 4, we conclude that the cost of debiasing algorithm is acceptable. Moreover, since the debiasing only depends on the number of nodes sampled, its time cost will be dwarfed by sampling on very large graphs. For example, sampling 512 nodes, the average batch sampling time for "LADIES", "LADIES + debiased", "LADIES + flat + debiased", are $8.3 \pm 0.1$, $11.7 \pm 0.2$, $11.7 \pm 0.5$ respectively on ogbn-arxiv while $83.4 \pm 1.3$ and $83.7 \pm 0.8$ and $83.5 \pm 0.6$ respectively on ogbn-products data. We remark that the node-wise sampling takes a significantly longer time because individually sampling from each row in the re-normalized Laplacian matrix (stored as a sparse matrix in implementation) leads to a large overhead cost.

We present the training time per batch for 2-layer and 3-layer GCNs with different methods in Tables 5 and Table 6 respectively. The time is presented in millisecond and averaged over 110 batches, where we discard the first 20 and the last 20 out of 150 total batches to disregard potential warm-up time for GPU. The other settings are kept the same as our experiments of accuracy evaluation in Table 1. We note that the timing on GPU is sensitive to the hardware and has a relatively large standard deviation.

As presented in Tables 5 and 6, our proposed methods have similar training time with LADIES due to the same propagation scheme of GCN with layer-wise sampling strategy. The VR-GCN generally shows superiority in prediction accuracy (see Table 1). However, it also takes a significantly longer time in training since its propagation involves using and updating historical activation.

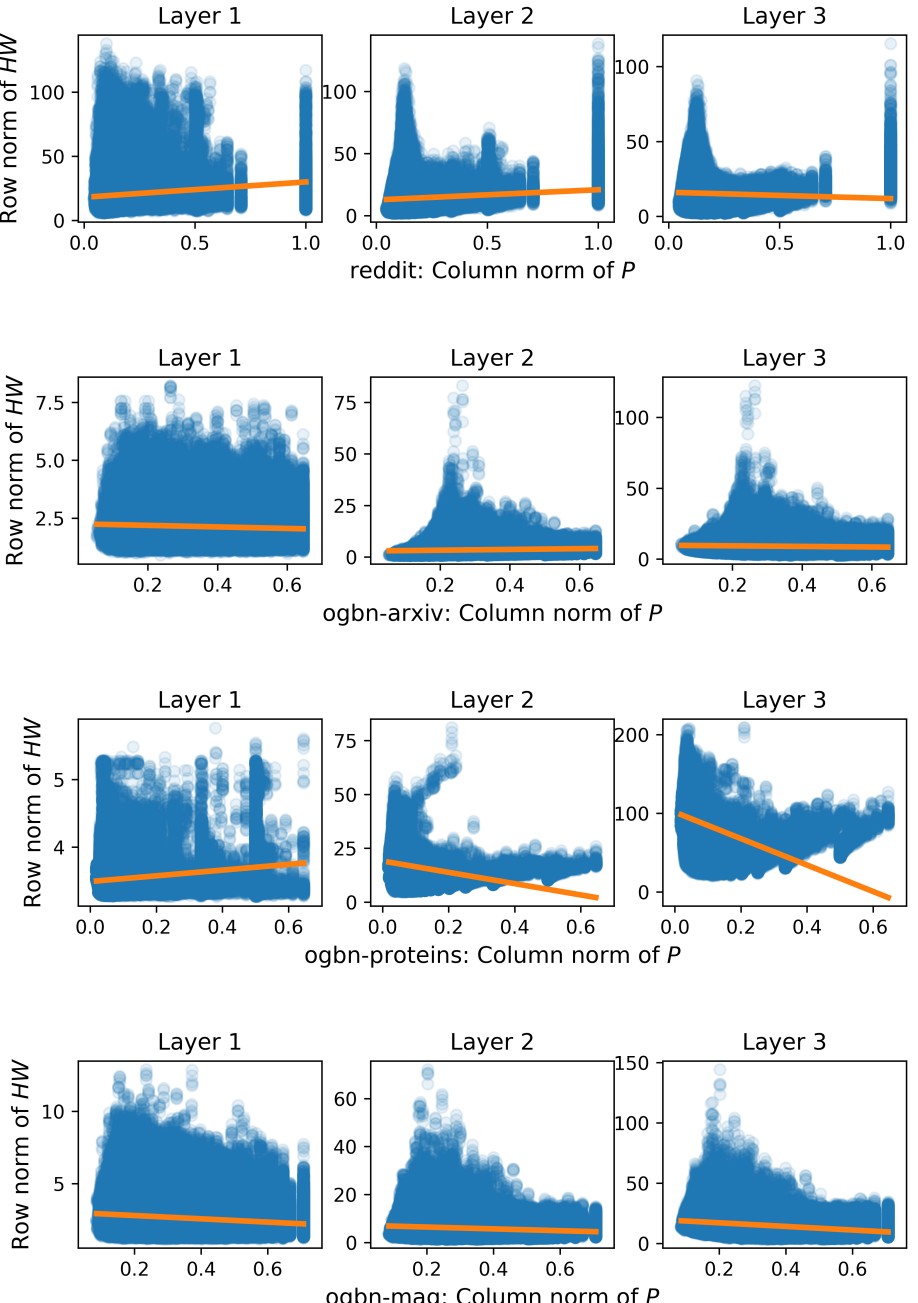

Figure 7: Regression on Reddit, ogbn-arxiv, ogbn-protein, ogbn-mag datasets. GCN is trained by LADIES + flat + debias sampler. The fitted regression line is in orange color.

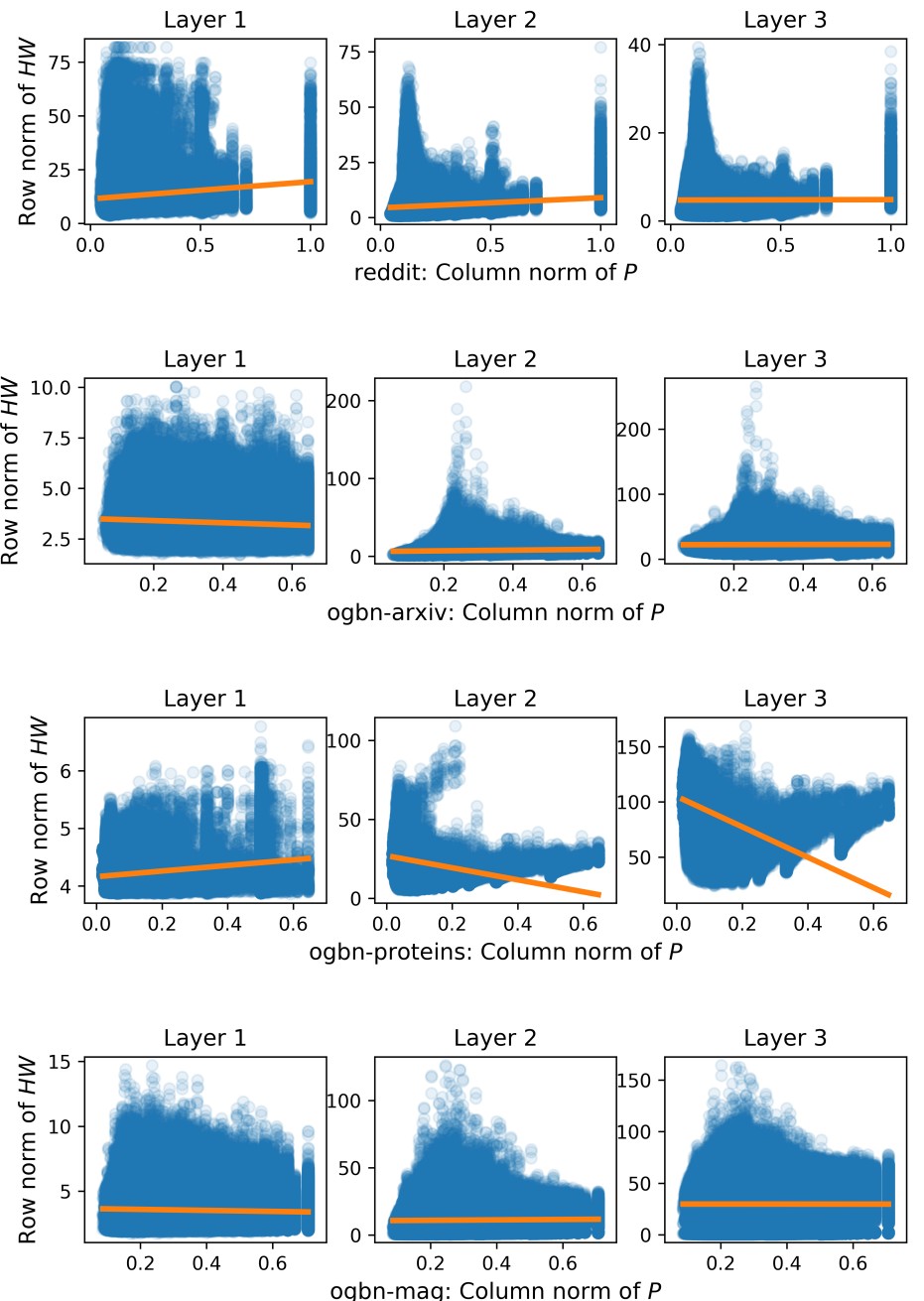

Figure 8: Regression on Reddit, ogbn-arxiv, ogbn-protein, ogbn-mag datasets. GCN is trained by LADIES. The fitted regression line is in orange color.

Table 4: Average sampling time (in milliseconds) per batch for layer-wise methods and vanilla node-wise method (GraphSAGE). 512 and 1024 (followed with the dataset name) indicate the number of node sampled for one layer. The "f" and "d" in "LADIES+f+d" denotes "flat" and "debiased" respectively.

| | FastGCN | LADIES | LADIES+f | LADIES+d | LADIES+f+d | Node-wise |
|---|---|---|---|---|---|---|
| Reddit (512) | $10.6 \pm 0.9$ | $10.9 \pm 0.3$ | $10.0 \pm 0.2$ | $13.1 \pm 0.3$ | $13.1 \pm 0.4$ | $632.5 \pm 4.3$ |
| Reddit (1024) | $10.0 \pm 0.6$ | $11.8 \pm 0.4$ | $10.4 \pm 0.1$ | $17.1 \pm 0.4$ | $16.1 \pm 0.6$ | $637.0 \pm 4.2$ |
| arxiv (512) | $4.2 \pm 0.1$ | $8.3 \pm 0.1$ | $7.8 \pm 0.1$ | $11.7 \pm 0.2$ | $11.7 \pm 0.5$ | $585.2 \pm 3.6$ |
| arxiv (1024) | $6.9 \pm 0.1$ | $9.7 \pm 0.1$ | $9.0 \pm 0.1$ | $17.2 \pm 0.3$ | $16.5 \pm 0.3$ | $585.6 \pm 3.1$ |
| mag (512) | $16.8 \pm 0.1$ | $27 \pm 0.1$ | $24.5 \pm 0.1$ | $30.0 \pm 0.03$ | $27.8 \pm 0.1$ | $1084.3 \pm 1.7$ |
| mag (1024) | $18.9 \pm 0.2$ | $28.6 \pm 0.2$ | $27.7 \pm 0.1$ | $36.0 \pm 0.2$ | $34.9 \pm 0.1$ | $1119 \pm 2.9$ |
| proteins (512) | $11.1 \pm 1$ | $11.2 \pm 0.3$ | $10 \pm 0.2$ | $13.6 \pm 0.2$ | $12.6 \pm 0.2$ | $830.9 \pm 5.3$ |
| proteins (1024) | $8.9 \pm 0.2$ | $12.4 \pm 0.1$ | $11.4 \pm 0.1$ | $18.9 \pm 0.2$ | $18.0 \pm 0.3$ | $804.2 \pm 4.6$ |
| products (512) | $54.8 \pm 0.7$ | $83.4 \pm 1.3$ | $80.3 \pm 0.4$ | $83.7 \pm 0.8$ | $83.5 \pm 0.6$ | $2795.4 \pm 4.7$ |
| products (1024) | $57.1 \pm 0.5$ | $80.8 \pm 0.8$ | $78.7 \pm 0.7$ | $87.0 \pm 0.6$ | $85.4 \pm 0.7$ | $2737.7 \pm 4.8$ |

Table 5: Average training time (in milliseconds) per batch for a 2-layer GCN.

| | Reddit | ogbn-arxiv | ogbn-mag | ogbn-proteins | ogbn-products |
|---|---|---|---|---|---|
| Full-batch | $372 \pm 21.5$ | $65.2 \pm 3.97$ | $72.3 \pm 8.25$ | $1702 \pm 67.0$ | $703 \pm 77.8$ |
| FastGCN | $9.47 \pm 1.21$ | $24.1 \pm 5.12$ | $23.3 \pm 6.27$ | $8.50 \pm 1.22$ | $7.43 \pm 1.04$ |
| FastGCN (2) | $9.47 \pm 1.25$ | $24.9 \pm 5.09$ | $20.8 \pm 5.71$ | $8.76 \pm 1.06$ | $8.34 \pm 1.11$ |
| LADIES | $7.95 \pm 1.03$ | $19.3 \pm 3.25$ | $16.7 \pm 4.54$ | $8.69 \pm 1.11$ | $10.3 \pm 1.24$ |
| w/ flat | $7.86 \pm 1.04$ | $16.0 \pm 2.37$ | $26.1 \pm 6.54$ | $9.00 \pm 1.21$ | $8.03 \pm 1.07$ |
| w/ debiased | $7.86 \pm 1.11$ | $19.1 \pm 3.45$ | $19.5 \pm 5.67$ | $8.21 \pm 1.04$ | $8.44 \pm 1.09$ |
| w/ flat & debiased | $8.01 \pm 1.09$ | $14.6 \pm 2.59$ | $20.5 \pm 5.62$ | $9.06 \pm 1.14$ | $8.11 \pm 1.09$ |
| LADIES (2) | $14.2 \pm 4.68$ | $21.0 \pm 4.00$ | $22.7 \pm 6.05$ | $8.83 \pm 1.12$ | $11.2 \pm 1.35$ |
| w/ flat | $8.70 \pm 1.13$ | $23.1 \pm 4.04$ | $21.9 \pm 5.65$ | $10.3 \pm 1.24$ | $13.1 \pm 1.52$ |
| w/ debiased | $8.75 \pm 1.15$ | $14.0 \pm 1.94$ | $16.3 \pm 4.64$ | $10.7 \pm 1.29$ | $8.54 \pm 1.09$ |
| w/ flat & debiased | $8.12 \pm 1.13$ | $21.0 \pm 3.60$ | $13.3 \pm 4.15$ | $12.5 \pm 1.41$ | $8.50 \pm 1.15$ |
| Node-wise (2) | $8.13 \pm 1.12$ | $22.0 \pm 3.64$ | $17.3 \pm 4.67$ | $9.50 \pm 1.29$ | $8.34 \pm 1.21$ |
| Node-wise (10) | $11.3 \pm 1.05$ | $19.7 \pm 2.84$ | $22.0 \pm 5.60$ | $10.3 \pm 1.29$ | $11.7 \pm 1.31$ |
| VR-GCN (2) | $153 \pm 17.3$ | $86.8 \pm 6.09$ | $106 \pm 12.4$ | $239 \pm 42.7$ | $88.5 \pm 2.51$ |
| VR-GCN (10) | $302 \pm 23.9$ | $104 \pm 8.47$ | $175 \pm 15.1$ | $360 \pm 45.6$ | $402 \pm 65.3$ |
| GraphSAINT | $8.23 \pm 1.14$ | $23.1 \pm 3.26$ | $20.4 \pm 5.98$ | $8.34 \pm 1.07$ | $8.26 \pm 1.11$ |

Table 6: Average training time (in milliseconds) per batch for a 3-layer GCN.

| | Reddit | ogbn-arxiv | ogbn-mag | ogbn-proteins | ogbn-products |
|---|---|---|---|---|---|
| Full-batch | $1042.8 \pm 30.1$ | $148 \pm 5.8$ | $352.1 \pm 11.5$ | $3312.3 \pm 82.1$ | $4490.7 \pm 102.6$ |
| FastGCN | $16.9 \pm 6.3$ | $30.8 \pm 4.3$ | $19 \pm 4.8$ | $13 \pm 1.3$ | $8.9 \pm 1.0$ |
| FastGCN (2) | $9.6 \pm 1.2$ | $27.4 \pm 4.6$ | $18.9 \pm 4.7$ | $9.8 \pm 1.2$ | $9.4 \pm 1.1$ |
| LADIES | $10.6 \pm 1.3$ | $27.6 \pm 3.7$ | $19.9 \pm 4.8$ | $11.1 \pm 1.3$ | $8.9 \pm 1.0$ |
| w/ flat | $10.1 \pm 1.1$ | $26.4 \pm 3.9$ | $12.1 \pm 2.4$ | $10.1 \pm 1.2$ | $9.9 \pm 1.0$ |
| w/ debiased | $10.1 \pm 1.2$ | $30 \pm 4.1$ | $22.5 \pm 5.5$ | $10.6 \pm 1.2$ | $10.6 \pm 1.0$ |
| w/ flat & debiased | $9.7 \pm 1.1$ | $27.1 \pm 3.6$ | $15.7 \pm 4.3$ | $10.9 \pm 1.2$ | $9.3 \pm 1.0$ |
| LADIES (2) | $10 \pm 1.2$ | $29 \pm 4.1$ | $19.2 \pm 5.1$ | $10.5 \pm 1.2$ | $10.2 \pm 1.1$ |
| w/ flat | $16.1 \pm 4.9$ | $27.2 \pm 3.7$ | $21.4 \pm 5.8$ | $10.4 \pm 1.1$ | $13.1 \pm 1.4$ |
| w/ debiased | $10.3 \pm 1.1$ | $24.7 \pm 3.1$ | $23.3 \pm 5.8$ | $10.5 \pm 1.2$ | $12.4 \pm 1.3$ |
| w/ flat & debiased | $10.7 \pm 1.1$ | $26.8 \pm 4.4$ | $24.8 \pm 6.4$ | $10.5 \pm 1.1$ | $9.9 \pm 1.1$ |
| Node-wise (2) | $10.9 \pm 1.3$ | $27 \pm 4.2$ | $17.2 \pm 4.3$ | $11.5 \pm 1.3$ | $11.5 \pm 1.1$ |
| Node-wise (10) | $77.8 \pm 12$ | $34.9 \pm 3.3$ | $52.4 \pm 6.7$ | $24.6 \pm 1.1$ | $50.4 \pm 1.1$ |
| VR-GCN (2) | $379.7 \pm 23.6$ | $154.1 \pm 12$ | $218.7 \pm 16.7$ | $428.9 \pm 53$ | $473.6 \pm 69.7$ |
| VR-GCN (10) | $858.1 \pm 32$ | $224.9 \pm 17.8$ | $488.1 \pm 34.7$ | $1618.7 \pm 67.3$ | $2075 \pm 112.8$ |
| GraphSAINT | $9.5 \pm 1.1$ | $22.6 \pm 3.2$ | $21.2 \pm 5.6$ | $11.2 \pm 1.3$ | $10.5 \pm 1.0$ |

### G.3 FINER VISUALIZATION OF FIGURE 3 AND FIGURE 4

To better differentiate the distinct sampling-based methods in Figure 3 and Figure 4, we provide Figure 9 to group the methods of the same type by the same color and demonstrate the trajectories of model convergence. We also provide Figure 10 to only compare the layer-wise sampling methods (FastGCN, LADIES, LADIES+flat, LADIES+debiasd, LADIES+flat+debiasd).

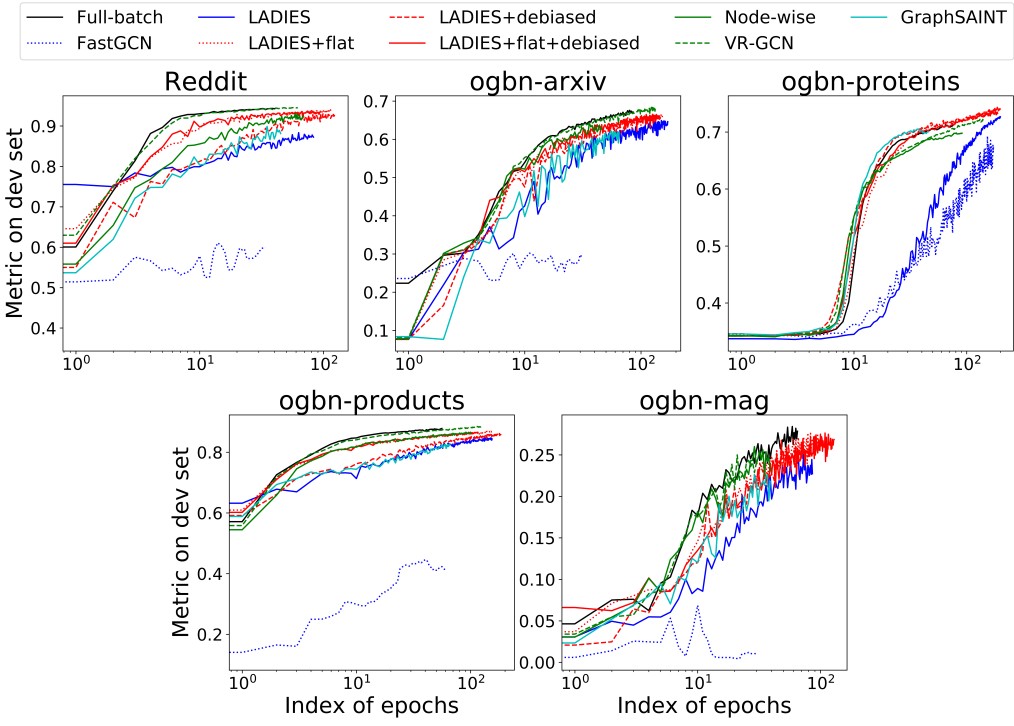

Figure 9: Metrics in each epoch.

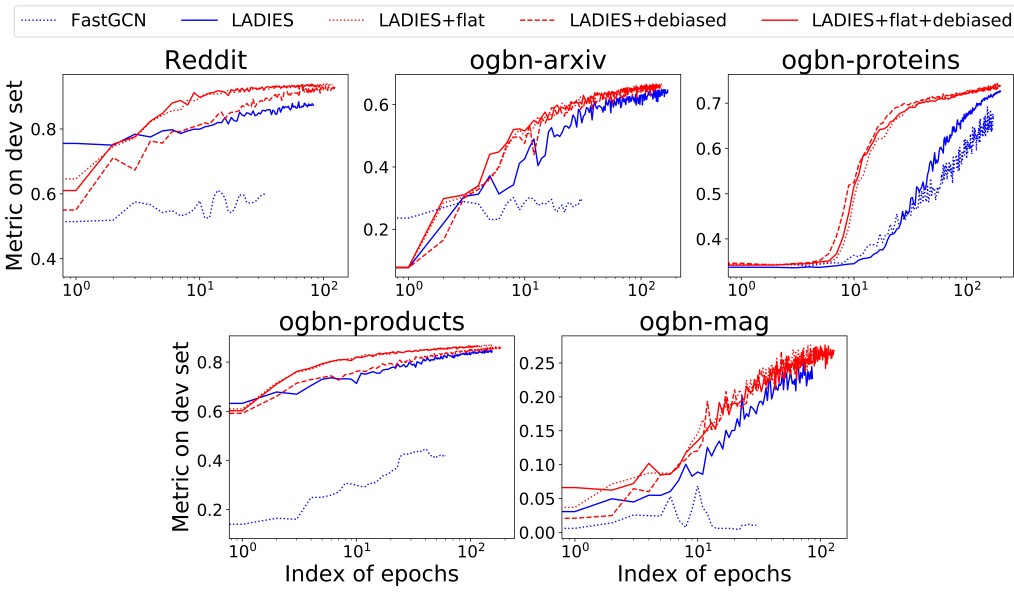

Figure 10: Metrics in each epoch.

