# OpenReview forum: "Revisiting Layer-wise Sampling in Fast Training for Graph Convolutional Networks"
_ICLR.cc/2022/Conference — ICLR 2022 Submitted_

### Official Review · Reviewer_xS1Y · 2021-11-01

**Correctness:** 3
**Technical Novelty And Significance:** 2
**Empirical Novelty And Significance:** 2
**Recommendation:** 3
**Confidence:** 5

**Main Review:**

## Strengths:
Overall, the presentation of this paper is clear. The experiments on the large-scale dataset look comprehensive.

## Weaknesses:
### 1. The theoretical contribution is limited.
This paper proposes a  sampling probability construction that abandons the hidden embedding matrix $H$ and the parameter matrix $W$. The core theoretical contribution is to prove that such construction has a better variance than that of `LADIES`. However, this result is weak:
- Limited scope: Lemma 1 only prove the new sampling probability is better than that of `LADIES`. The superiority of new sampling probability over the other layer-wise sampling method, such as FastGCN and AS-GCN, is still unclear. It’s hard to say the new sampling probability is superior to the other layer-wise sampling methods from the theoretical perspective.
- Strong assumption: To complete this proof, the authors give a very strong assumption (Assumption 1) without any justification of the existence and rationality.  There is no in-depth discussion about why and when this assumption is held in practice. For instance, Since the assumption ($C_1/C_2^2 \leq 1$) is highly related to $HW$,  does this assumption still hold during the training process? Therefore, such a strong assumption makes Lemma  1 impractical and its theoretical value to the real application is limited.


### 2. The experimental results are weak.
- The important layer-wise baseline `ASGCN` is missing.

- On Reddit dataset, the results of several baselines are significantly lower than the results in the existing literature. For example, the original F1-score of `GraphSAINT` is  0.966, while in this paper, the F1-score is only 0.8947. I guess this inconsistency may due to the different data splitting. However, the author doesn’t give the details of the data splitting setting, nor gives the reason why does not use the public splitting.

-  As shown in Table 2, this paper only applies the new sampling probability and debiasing method to `LADIES` and observes an improvement. However, even with such a new sampling probability and debiasing approach, the performance is still worse than the other baselines.  Additionally, in Figure 3, the curve of  `LADIES+flat+debiased` is lower than that of other baselines. It’s hard to say the new method is better than that of other methods.

-  There is no time comparison,  especially the sampling time comparison,  across different methods. Since the debiasing operation may introduce the additional computation overhead, I’m curious about such overhead in practice.


### 3. Minor issue
- In Figure 1, why `FastGCN` achieves a better approximation on `ogbn-proteins` than `LADIES`?
- In my understanding, the sampling probability and sampling method are two core parts of a layer-wise sampling method. Therefore, it is improper to treat the new approach as the extension of `LADIES` and use the notation `+flat+debiased` to represent the new approach.

## Reference:
[`FastGCN`] FastGCN: Fast Learning with Graph Convolutional Networks via Importance Sampling

[`ASGCN`] Adaptive Sampling Towards Fast Graph Representation

[`LADIES`] Layer-Dependent Importance Sampling for Training Deep and Large Graph Convolutional Networks

[`GraphSAINT`] GraphSAINT: Graph Sampling Based Inductive Learning Method


**Summary Of The Paper:**

This paper introduced a new layer-wise sampling probability as well as a new debiasing sampling algorithm to obtain a better approximation accuracy.

**Summary Of The Review:**

Overall, I think that this paper is not ready to be accepted. The details are elaborated in main review.

---

> ### Author Response · Authors · 2021-11-18
> **Author Response of Paper2563**
>
> We appreciate your comments and summaries. We hope our following responses properly address all your concerns.
>
> ### Weakness 1 regarding theoretical contribution
>
> We hope to argue that our theoretical contribution is comparable to the previous works. To the best of our knowledge, our paper is the first to reconsider and theoretically analyze the implicit proportionality assumption in LADIES and FastGCN. In addition, our theoretical contribution includes the analysis of the bias issue in the implementation of sampling in previous work and the proposition of a theoretically guaranteed debiasing algorithm.
>
> For your concern regarding “limited scope”, we remark that only models under the same framework can be theoretically comparable.
> - Specifically, the variance of FastGCN and LADIES (derived in Appendix B of LADIES) depends on many factors, including graph properties, choices of nodes in a batch, constants in their Assumption 1,2, etc. Thus although LADIES is a direct improvement over FastGCN they do not explicitly claim that LADIES always outperforms FastGCN.
> - Similarly, the variance of AS-GCN depends on how well their adaptive structure captures the information of $||HW||$, which is intractable in existing theoretical analysis. This makes the variance comparison to AS-GCN difficult. We follow the same logic as in LADIES and do not compare AS-GCN in our work.
>
> For your concern regarding “strong assumption”, we have provided some discussion at the end of Page 5 and in Appendix F. Besides:
> - We hope to argue Assumption 1 seems strong but it is justified by empirical evidence. As a recap, in Section 4.4 and Appendix F we remark that the assumption implies the distribution of column norms, $||(QP)^{[i]}||$’s, is long-tailed, and the row norms, $||(HW)_{[i]}||$’s, has a small variation. The latter claim is empirically shown by Figure 6 in Appendix C.4, while the outliers observed in the first column of figure 6 empirically support the long-tailed distribution of $||(QP)^{[i]}||$.
> - The empirical results also imply that at the stable convergent stage (the last stage during the training process), our Assumption 1 tends to hold.
> - We remark that the implicit proportionality assumption used by LADIES and FastGCN is even stronger.
>
>
> ### Weakness 2 regarding experimental results
>
> For AS-GCN, we have provided discussions in Sections 1.1 and 4.1. AS-GCN is an accurate model but GraphSAINT (Section 5.1) also points out the large time cost of AS-GCN: ‘the sampler of AS-GCN is expensive to execute, making its overall training time even longer than vanilla GCN’. We have commented in the paper that AS-GCN is irrelevant to the main focus of our paper, the improvement of layer-wise sampling without sacrificing efficiency. We guess that it is also the reason why AS-GCN is not compared in LADIES. Furthermore, we have compared our methods with VR-GCN, which follows a similar philosophy as in AS-GCN to model $HW$.
>
> For the concern on data split and the performance of GraphSAINT, we indeed follow the default data splitting of Reddit (c.f. Table 2). The accuracy gap comes from the “node budget” for GraphSAINT (the sub-graph size for Reddit is 8,000 in the paper of GraphSAINT, while it is 512 in our experiments for a fair comparison with other methods).
>
> For the concern that our proposed methods did not achieve the best performance on some tasks, we reiterate that our paper pays more attention to layer-wise sampling scheme as well as the trade-off between accuracy and efficiency, not to show SOTA over all the other non-layer-wise sampling methods. In particular, for the performance in Table 1 (we assume that you are referring to Table 1 as Table 2), we argue that methods with orders of magnitude more computational cost can surely achieve better performance on some tasks. For example, VR-GCN requires much more additional costs to store and use historical activations. (see newly added Table 5,6 in Appendix G.2 for the cost). Similarly, for Figure 3, we remark that it is not surprising the methods with larger cost can converge faster in terms of the number of epochs.
>
> For the comparison of runtime for sampling and training, we summarize the results in our general response.

---

> > ### Author Response · Authors · 2021-11-18
> > **Author Response of Paper2563 - Minor Issues**
> >
> > ### 1: The matrix approximation error of FastGCN and LADIES on ogbn-proteins
> >
> > The discussion is spared since those two methods are existing methods, not the method of interest. To address your concern, we would like to briefly discuss the phenomenon here and add the discussion to our paper’s next version.
> >
> > Indeed, a relevant phenomenon can be observed in Figure 10 (or Figure 3) that LADIES converges slower than FastGCN at the beginning on ogbn-proteins. We speculate the poorer performance of LADIES is caused by the dense connection in ogbn-proteins (597 degrees on average).
> > In principle, the LADIES does not guarantee a smaller variance than FastGCN on the dense graphs. The layer-dependent sampling scheme (LADIES) is proposed to address the sparse connection issue (see Section 3 in LADIES) for FastGCN, which usually occurs in very sparse graphs.
> > Theoretically, the variance upper bound of LADIES (they provide the full derivation in Appendix B of their paper) is proportional to “the average number of nodes that are connected to the nodes sampled in the training batch.”
> > We guess LADIES tends to have a larger bias than FastGCN since the bias caused by weighted random sampling is larger for a smaller "population" size. The "population" of FastGCN is all the nodes while that of LADIES is all the neighbors in the current layer. The matrix approximation error comes from both variance and bias. The bias of LADIES seems non-negligible since later we show that our debiasing method significantly reduces the matrix approximation error of LADIES (see the curve of “LADIES+debias” in Figure 1).
> >
> > ### 2: The names of our proposed methods
> >
> > We appreciate your suggestion and would reconsider the proper names in the next version.

---

### Official Review · Reviewer_cNek · 2021-11-02

**Correctness:** 3
**Technical Novelty And Significance:** 3
**Empirical Novelty And Significance:** 3
**Recommendation:** 6
**Confidence:** 2

**Main Review:**

**Strengths**

(1) This paper is the first to address the issue of strong assumption, which induces a sub-optimal solution.

(2) Overall, the paper is well written. In particular, the Notations and Preliminaries section is well discussed.

(3) This paper provides comprehensive experiments to show the effectiveness of the proposed learning schemes.

**Weakness**

(1) The time complexity is an important issue of sampling schemes of graph neural networks. However, there is no empirical study of the efficiency of the proposed method.

(2) What's the main intuition behind the sampling probabilities in Equation (3)?



**Summary Of The Paper:**

This paper revisits layer-wise sampling methods of graph neural networks by addressing issues such as sub-optimal sampling probabilities and the approximation bias due to the usage of sampling without replacement. This paper also presents a new metric to evaluate the performance of the sampling strategy. Through comprehensive experiments, the authors validate the effectiveness of their proposed methods.

**Summary Of The Review:**

Overall, I vote for accepting. The paper well presents their proposed method with motivation.

---

> ### Author Response · Authors · 2021-11-18
> **Author Response of Paper2563**
>
> We thank you for your interest in our paper and appreciate all your helpful feedback. For your questions, we respond as follows.
>
> ### Weakness 1: The time complexity is an important issue of sampling schemes of graph neural networks. However, there is no empirical study of the efficiency of the proposed method.
>
> Please refer to our general response for more details.
>
>
>
> ### Weakness 2: What's the main intuition behind the sampling probabilities in Equation (3)?
>
> We have briefly discussed intuition in Section 4. The main idea of Equation (3) is that we discard the “proportionality assumption” by FastGCN and LADIES and instead consider the $\|(HW)_{[i]}\|$ as invariant to $i$, so that in the proposed sampling probability  (see page 4)
>
> $$p_i \propto ||(HW)_{[i]}|| \cdot ||(Q P)^{[i]}||.$$
>
> We provide more details about the intuition as follows:
> - The ideal importance sampling probabilities is $p_i \propto ||(HW)_{[i]}|| \cdot ||(Q P)^{[i]}||$. However, as discussed in Section 4.1, we tend to not access $HW$ to avoid the heavy computational cost and to attain the separation between sampling and training.
> - FastGCN and LADIES assume that $||(HW)_{[i]}||$ is proportional to the corresponding $||(Q P)^{[i]}||$. Therefore they make $p_i$ proportional to $||(Q P)^{[i]}||^2$. However, our regression experiments (Figure 2) and matrix approximation error (Figure 1) show that their sampling probabilities are sub-optimal.
> - Instead of the proportionality assumption, we assume $||(HW)_{[i]}||$ distributed uniformly over $i$. This leads to our conservative sampling probabilities in Equation (3).

---

### Official Review · Reviewer_YZaF · 2021-11-03

**Correctness:** 3
**Technical Novelty And Significance:** 3
**Empirical Novelty And Significance:** 3
**Recommendation:** 5
**Confidence:** 3

**Main Review:**

Strength:

The summary of related work is good and clear. The authors provide theoretically analysis which can be verified by their experiments.

Weakness&Advice:

1. Need citations after “the phenomenon of ‘neighbor explosion’ ”.

2. What is the running time of your method compared to the existing layer-wise sampling methods and baseline methods?

3. What is the regression result of your proposed sampling method with regard to $H^{(l)}W^{(l)}$

4. The visualization in Figure 3 needs to be improved. The font size is too small and the curves are messy.

**Summary Of The Paper:**

The authors propose remedies to address sub-optimal sampling probabilities and the approximation bias induced by sampling without replacement  issues in layer-wise sampling methods of GNNs.

**Summary Of The Review:**

The novelty and difference with existing works need to be emphasized. I'll consider raise my score if the authors can address my concerns properly.

---

> ### Author Response · Authors · 2021-11-18
> **Author Response of Paper2563**
>
> We thank you for all your valuable comments. We hope the summary in our general response and our following replies answer your questions.
>
> ### 1: Need citations after “the phenomenon of ‘neighbor explosion’ ”.
>
> Thanks for the suggestions. We have updated the first paragraph in our recent rebuttal revision accordingly.
>
> ### 2: What is the running time of your method compared to the existing layer-wise sampling methods and baseline methods?
>
> Please refer to our general response for more details.
>
> ### 3: What is the regression result of your proposed sampling method with regard to $H(l)W(l)$
>
> We present additional regression experiments in Appendix G.1. Figure 7 shows the regression lines with our methods, which are very close to the original results in Figure 2 and 5. Similarly, many regression lines show a negative slope. These results also demonstrate the violation of the “proportionality assumption” used by FastGCN and LADIES.
>
> ### 4: The visualization in Figure 3 needs to be improved. The font size is too small and the curves are messy.
>
> Thanks for the suggestions. We have provided a finer visualization of Figure 3 and Figure 4 in Appendix G.3, and another new figure which restricts the comparison within layer-wise methods.

---

### Official Review · Reviewer_Rhqb · 2021-11-06

**Correctness:** 3
**Technical Novelty And Significance:** 2
**Empirical Novelty And Significance:** 2
**Recommendation:** 5
**Confidence:** 4

**Main Review:**

Highlights:
- Paper proposes uniform distribution over nodes in the batch when training
  - Looks at the empirical Frobenius norm (matrix approximation error) of the difference between a non-sampling and sampling batch for the first layer outputs. Empirical results show a reduction when using uniform distribution.
  - Applies regression analysis between the $\ell_2$ norm of one row $i$ in $\boldsymbol{H}^{(l)}\boldsymbol{W}^{(l)}$ and corresponding column $i$ in $\boldsymbol{P}$ for different layers $l$. Empirical observation of instability and potential negative correlation demonstrates that the LADIES assumption of $||(HW)_{[i]}|| \propto ||P^{[i]}||$ is violated.
  - Theoretical analysis is adapted from LADIES (Zou et al. 2019) result. Assumption here is when the assumption in LADIES is not met ( $||\boldsymbol{H}\boldsymbol{W}_{(i)}|| \not\propto$ corresponding $\ell_2$ norm of column $(\boldsymbol{Q}\boldsymbol{P})^{(i)}$), then the proposed method will work well.

- Paper proposes algorithm to reduce bias
  - Provides analysis of non-uniform sampling without replacement in FastGCN and LADIES
  - Gives simple recursive weighted average adjustment to sampling without replacement and runtime complexity

Strength:
- Empirical observations and analyses are interesting
- Results show improvement in prediction accuracy

Weakness:
- Theoretical analysis needs more rigor
- Analysis and comments as it relates to network structure would be helpful
- Dataset dependent and limited
- Incremental improvement to LADIES

Misc:

- Section 4.1: second paragraph, second line, "improves"
- Section 4.4: extra $||$ in line 4
- The way the citations are applied can be improved (Section 1: first paragraph, first sentence)



**Summary Of The Paper:**

This paper studies existing sampling schemes employed in training graph neural network architectures (e.g., FastGCN [Chen et al.] and LADIES [Zon et al.]) that are improvements to graph convolutional networks (GCN) [Kipf & Welling 2017] from a matrix approximation perspective (looking a Frobenius norm).  The paper focuses on layer-wise sampling (cf. node and subgraph sampling) and observes that there are two drawbacks in common practice layer-wise sampling. First drawback is current probabilities (probability distributions) used for sampling are sub-optimal. The reason the paper gives is that a core assumption made by certain GCN schemes such as FastGCN and LADIES do not hold in datasets such as Reddit and OGB. Second drawback is the implementations of these schemes slightly deviate from their respective theoretical results. The implementations use sampling without replacement, and thus introduces bias.  To this, the paper presents new sampling probabilities (uniform proposal distribution), and an algorithm to reduce the bias. With these adjustments, training of GCN converges faster and potentially leads to higher prediction accuracy (for node prediction). Theoretical analysis is presented and experiments are conducted on common benchmarks (Reddit, ogbn-{arxiv,proteins,products,mag}).


**Summary Of The Review:**

Although the paper presents interesting empirical observations and analysis, mainly, it is dataset centric. The theoretical justification of uniform distribution is weak -- again, using uniform distribution comes down to data dependency.

---

> ### Author Response · Authors · 2021-11-18
> **Author Response of Paper2563**
>
> We appreciate your valuable summaries of the importance of our work. We hope your concerns can be addressed by the following responses.
>
> ### Weakness 1: Theoretical analysis needs more rigor
>
> - First, we hope to argue Assumption 1 seems strong but it is justified by empirical evidence.
> As a recap, in Section 4.4 and Appendix F we remark that the assumption implies the distribution of column norms, $||(QP)^{[i]}||$’s, is long-tailed, and the row norms, $||(HW)_{[i]}||$’s, has a small variation. The latter claim is empirically shown by Figure 6 in Appendix C.4, while the outliers observed in the first column of figure 6 empirically support the long-tailed distribution of $||(QP)^{[i]}||$.
>
> - Secondly, FastGCN and LADIES do not provide theoretical or empirical evidence to support their even stronger ‘proportionality assumption’. That is the reason why they dismiss the sub-optimality of their sampling probability. In this sense, our result has improved upon the previous literature.
>
> ### Weakness 2: data dependent and limited
>
> It is common that most literature on sampling-based training of graph neural networks is mainly methodology-driven and does not focus on developing rigorous theory. The evaluation of most existing literature is based on open benchmarks, and it is inevitable that we have a data-dependent evaluation component under such a topic. Meanwhile, our numerical evaluation is thorough, considering that Reddit and multiple OGB datasets are popular in recent literature and much larger than most datasets used in the early work (e.g. FastGCN, LADIES). We believe that these graph datasets are representative of current mainstream node classification tasks.
>
> To provide details under this context, as mentioned in Section 3, the Reddit data is used by lots of previous works related to the sampling-based fast training for GCN, including FastGCN, LADIES, GraphSAINT, GraphSAGE, and VR-GCN, etc. In addition, OGB datasets are very popular and have been cited more than 300 times since being proposed in 2020. These datasets exhibit highly diverse graph statistics (Hu et al. 2020), as summarized in Table 2 in Appendix B. The ogbn-proteins dataset is a dense network with 597 average degrees, which is 10 times more than that of other datasets; the ogbn-products dataset has more than 2 million nodes and uses an $8\%/ 2\% / 90\%$ splitting ratio for training/validation/testing respectively, which challenges scalability to large-scale graphs and out-of-distribution generalization.
>
>
>
> ### Weakness 3: Analysis and comments as it relates to network structure would be helpful
>
> We understand your concern as the lack of connection between the sampling scheme and the network structure. We do appreciate any further clarification if we fail to get your point.
>
> - First, we remark that we have reviewed the connection between GCN and sampling-based training. The propagation formula of GCN,  $H^{(l+1)} = \sigma (H^{(l)} P W^{(l)})$ (see E.q. (1)), is different from that of common convolutional network, i.e. $H^{(l+1)} = \sigma (H^{(l)} W^{(l)})$. The involvement of matrix $P$, i.e. the renormalized Laplacian matrix, results in the neighbor explosion (expansion) issue (c.f. FastGCN, GraphSAINT). This makes straightforward SGD training inefficient and motivates the sampling-based stochastic training methods.
>
> - Secondly, most sampling based methods, including all the other methods we have compared, restrict their analysis in each layer. Since the focus of this paper is to identify the sub-optimality in previous layer-wise sampling methods, we follow their settings and do not dive into further discussion related to network structures.
>
> We appreciate your advice and would add the discussion to the next version of our paper.
>
>
> ### Weakness 4: incremental improvement to LADIES
>
> We hope to argue that our contribution is more than incremental improvements.
>
> - First, technically speaking, FastGCN, LADIES, and our proposed methods are all applications of randomized sketching theory in matrix approximation (c.f. Appendix E). For the importance sampling used in GCN training, we contribute to two aspects, the sampling probabilities and weighted random sampling (WRS) algorithm.
>
> - Secondly, our proposed debiasing algorithm is not limited to layer-wise sampling. It can be applied to non-uniform node-wise sampling schemes and a large family of machine learning algorithms using WRS (e.g. variance reduction in SGD).
>
> ### Misc
> We appreciate your remarks on the paper writing. We have corrected the typos and updated the usage of citations in the recent rebuttal revision.

---

### Author Response · Authors · 2021-11-18
**General Response of Paper2563**

We thank the reviewers for taking the time to review our paper and for their insightful comments. We manage to address all the concerns by the following individual response and the new materials in Appendix G (added to the rebuttal revision). Before we start to individually reply to the posts, we first summarize our contributions as follows to help recall the context.

- Reconsider and examine (see Figures 2 and 5) the implicit “proportionality assumption” in LADIES and FastGCN, and propose a new sampling mechanism to resolve the issue.
- Suggest a potential assumption (Assumption 1) under which our new importance sampling probabilities are proven to enjoy a lower variance than LADIES.
- Recognize the bias in the implementation of layer-wise sampling and propose our theoretically guaranteed debiasing algorithm (see Appendix A).

---

> ### Author Response · Authors · 2021-11-18
> **General Response of Paper2563 - Discussion on runtime**
>
> As some reviewers are interested in the empirical runtime of our methods, we leave a long discussion as follows:
>
> ​​The computational cost can conceptually be decomposed into two parts: the cost for sampling and the cost for training. We usually care more about the training cost, because the sampling procedure of FastGCN, LADIES, and our methods can be separated from the training procedure. Practically it is feasible to prepare all the sampled Laplacian matrices before the start of training on the GPU. As discussed in Section 4, this is considered as one advantage of common layer-wise sampling methods. However, it does not hold for AS-GCN and VR-GCN, which have to use recent information during the training.
>
> Additional experimental results on runtime presented in Appendix G.2 are organized as follows. Table 4 presents the batch sampling for 1-layer GCN on 5 datasets with layer-wise sampling methods and vanilla node-wise sampling method. Table 5 and Table 6 present the training time of all the methods for 2-layer GCN and 3-layer GCN respectively.
>
> We copy some representative results here. The following table shows the sampling time per batch for 1-layer GCN with 512 or 1024 sampled nodes on ogbn-arxiv and ogbn-products.  By comparing the sampling time we claim that the cost of debiasing algorithm is acceptable. Moreover, since the debiasing only depends on the number of nodes sampled, its time cost will be dwarfed by sampling on very large graphs, such as the ogbn-products. As a side note, we remark the inefficiency of node-wise sampling comes from the overhead cost in the sparse matrix (see discussions in Appendix G.2).
>
> | Time (ms per batch) 	| \# Nodes 	| FastGCN        	| LADIES         	| LADIES+flat    	| LADIES+debiased       	| LADIES+flat+debiased     	| Node-wise        	|
> |---------------------	|----------	|----------------	|----------------	|----------------	|----------------	|----------------	|------------------	|
> | ogbn-arxiv          	| 512      	| 4.2 $\pm$ 0.1  	| 8.3 $\pm$ 0.1  	| 7.8 $\pm$ 0.1  	| 11.7 $\pm$ 0.2 	| 11.7 $\pm$ 0.5 	| 585.2 $\pm$ 3.6  	|
> |                     	| 1024     	| 6.9 $\pm$ 0.1  	| 9.7 $\pm$ 0.1  	| 9.0 $\pm$ 0.1    	| 17.2 $\pm$ 0.3 	| 16.5 $\pm$ 0.3 	| 585.6 $\pm$ 3.1  	|
> | ogbn-products       	| 512      	| 54.8 $\pm$ 0.7 	| 83.4 $\pm$ 1.3 	| 80.3 $\pm$ 0.4 	| 83.7 $\pm$ 0.8 	| 83.5 $\pm$ 0.6 	| 2795.4 $\pm$ 4.7 	|
> |                     	| 1024     	| 57.1 $\pm$ 0.5 	| 80.8 $\pm$ 0.8 	| 78.7 $\pm$ 0.7 	| 87.0 $\pm$ 0.6   	| 85.4 $\pm$ 0.7 	| 2737.7 $\pm$ 4.8 	|
>
>
> The next table shows the training time of a 2-layer GCN with different methods. Our proposed methods have similar training time with LADIES due to the same propagation scheme of GCN as a layer-wise sampling strategy. We remark that though VR-GCN shows general superiority in prediction accuracy (see Table 1), it also takes a significantly longer time in training since its propagation involves applying and updating historical activation. We also note that the timing on GPU is sensitive to the hardware and has a relatively large standard deviation.
>
>
>
> |                     	| Reddit          	| ogbn-arxiv      	| ogbn-proteins    	| ogbn-products   	|
> |---------------------	|-----------------	|-----------------	|------------------	|-----------------	|
> | Full-batch          	| 372 $\pm$ 21.5  	| 65.2 $\pm$ 3.97 	| 1702 $\pm$ 67.0  	| 703 $\pm$ 77.8  	|
> | FastGCN             	| 9.47 $\pm$ 1.21 	| 24.1 $\pm$ 5.12 	| 8.50 $\pm$ 1.22  	| 7.43 $\pm$ 1.04 	|
> | FastGCN (2)         	| 9.47 $\pm$ 1.25 	| 24.9 $\pm$ 5.09 	| 8.76 $\pm$ 1.06  	| 8.34 $\pm$ 1.11 	|
> | LADIES              	| 7.95 $\pm$ 1.03 	| 19.3 $\pm$ 3.25 	| 8.69 $\pm$ 1.11  	| 10.3 $\pm$ 1.24 	|
> | w/ flat             	| 7.86 $\pm$ 1.04 	| 16.0 $\pm$ 2.37 	| 9.00 $\pm$ 1.21  	| 8.03 $\pm$ 1.07 	|
> | w/ debiased         	| 7.86 $\pm$ 1.11 	| 19.1 $\pm$ 3.45 	| 8.21 $\pm$ 1.04  	| 8.44 $\pm$ 1.09 	|
> | w/ flat \& debiased 	| 8.01 $\pm$ 1.09 	| 14.6 $\pm$ 2.59 	| 9.06 $\pm$ 1.14  	| 8.11 $\pm$ 1.09 	|
> | LADIES (2)          	| 14.2 $\pm$ 4.68 	| 21.0 $\pm$ 4.00 	| 8.83 $\pm$ 1.12  	| 11.2 $\pm$ 1.35 	|
> | w/ flat             	| 8.70 $\pm$ 1.13 	| 23.1 $\pm$ 4.04 	| 10.3 $\pm$ 1.24  	| 13.1 $\pm$ 1.52 	|
> | w/ debiased         	| 8.75 $\pm$ 1.15 	| 14.0 $\pm$ 1.94 	| 10.7 $\pm$  1.29 	| 8.54 $\pm$ 1.09 	|
> | w/ flat \& debiased 	| 8.12 $\pm$ 1.13 	| 21.0 $\pm$ 3.60 	| 12.5 $\pm$ 1.41  	| 8.50 $\pm$ 1.15 	|
> | Node-wise (2)       	| 8.13 $\pm$ 1.12 	| 22.0 $\pm$ 3.64 	| 9.50 $\pm$ 1.29  	| 8.34 $\pm$ 1.21 	|
> | Node-wise (10)      	| 11.3 $\pm$ 1.05 	| 19.7 $\pm$ 2.84 	| 10.3 $\pm$ 1.29  	| 11.7 $\pm$ 1.31 	|
> | VR-GCN (2)          	| 153 $\pm$ 17.3  	| 86.8 $\pm$ 6.09 	| 239 $\pm$ 42.7   	| 88.5 $\pm$ 2.51 	|
> | VR-GCN (10)         	| 302 $\pm$ 23.9  	| 104 $\pm$ 8.47  	| 360 $\pm$ 45.6   	| 402 $\pm$ 65.3  	|
> | GraphSAINT          	| 8.23 $\pm$ 1.14 	| 23.1 $\pm$ 3.26 	| 8.34 $\pm$ 1.07  	| 8.26 $\pm$ 1.11 	|

---

### Decision · Program_Chairs · 2022-01-20

**Decision:**

Reject

**Comment:**

The reviewers think the proposed method is well motivated and interesting. However, the novelty needs to be improved. At the moment, the paper seems to be a minor improvement over existing works.